behaviour/environmental science

*Geospiza fortis*, *Geospiza fuliginosis*, foraging, human influences, urbanization, Galapagos

**Author for correspondence:**
K. M. Gotanda
e-mail: kg419@cam.ac.uk

# Darwin's small and medium ground finches might have taste preferences, but not for human foods

D. Lever[1], L. V. Rush[2], R. Thorogood[1,3,4] and
K. M. Gotanda[1,5,6]

[1]Department of Zoology, University of Cambridge, Downing Street, Cambridge CB2 3EJ, UK
[2]Department of Geology, Laurentian University, 935 Ramsey Lake Rd, Sudbury, Ontario P3E 2C6, Canada
[3]Helsinki Institute of Life Science (HiLIFE), and [4]Research Program in Organismal and Evolutionary Biology, Faculty of Biological and Environmental Sciences, University of Helsinki, Helsinki 00014, Finland
[5]Départment de Biologie, Université de Sherbrooke, 2500, boul de l'Université, Sherbrooke, Québec J1K 2R1, Canada
[6]Department of Biological Sciences, Brock University, 1812 Sir Isaac Brock Way, St Catharine's, Ontario L2S 3A1, Canada

RT, 0000-0001-5010-2177; KMG, 0000-0002-3666-0700

Urbanization is rapidly changing ecological niches. On the inhabited Galapagos Islands, Darwin's finches consume human-introduced foods preferentially; however, it remains unclear why. Here, we presented pastry with flavour profiles typical of human foods (oily, salty and sweet) to small ground finches (*Geospiza fuliginosa*) and medium ground finches (*Geospiza fortis*) to test if latent taste preferences might drive the selection of human foods. If human food flavours were consumed more than a neutral or bitter control only at sites with human foods, then we predicted tastes were acquired after urbanization; however, if no site differences were found then this would indicate latent taste preferences. Contrary to both predictions, we found little evidence that human food flavours were preferred compared with control flavours at any site. Instead, finches showed a weak aversion to oily foods, but only at remote (no human foods present) sites. This was further supported by behavioural responses, with beak-wiping occurring more often at remote sites after finches tasted flavours associated with human foods. Our results suggest, therefore, that while Darwin's finches regularly exposed to human foods might have acquired a tolerance to human food flavours, latent taste preferences are unlikely to have played a major role in their dietary response to increased urbanization.

# 1. Introduction

Human behaviour is now recognized to be a strong driver of local adaptation and differences among populations of animals [1,2]. Urbanization [3–5], for example, can have profound effects on foraging because humans often introduce novel foods to the surrounding environment either intentionally (e.g. via garden bird feeding [6–8]) or unintentionally (e.g. by planting ornamental, invasive plants [9,10] or food waste [11,12]). This changes the diversity and availability of food items and generates different foraging landscapes from those in which most animals evolved [13–15]. However, organisms' responses to these altered niches vary, with some birds, for example, not only adapting more readily to incorporate human foods into their diet, but even preferentially consuming them over native food sources [14,16–19]. These differences in whether, and how, populations and species use human foods can then have consequences for local adaptation [7,20] and potentially affect species divergence [21]. Taste preferences can also have implications in conservation, mediating the potential for invasive species to establish [22,23], for example, or making native species vulnerable to accidental poisoning [24]. Furthermore, the gut microbiome, which is increasingly recognized to affect a suite of physiological, immune and cognitive functions in wild non-human animals [25,26], can also vary with the consumption of human foods [27–29]. Nevertheless, understanding why some species in urban areas shift their diets to preferentially forage on human foods remains unclear.

Taste plays an important role in foraging as it allows individuals to detect nutritious versus unprofitable substances in food items [30,31]. Although taste was long assumed to be of little importance for birds, they have sophisticated sensory adaptations with taste receptors identified for bitter [30,32,33], sweet [30] and salt [34] flavours. Furthermore, experiments have demonstrated that birds can use bitter tastes to avoid toxic foods [32,34,35], use sweet tastes to detect sugars [30,34,36–39] indicating high caloric content [40] and use salty tastes to detect salts [39,41,42] indicative of necessary minerals and proteins [43]. Preferences for certain tastes can therefore evolve when flavours are linked to food quality and nutrient content [30]. For example, omnivorous and frugivorous birds are able to detect sugars at low levels and prefer sweet tastes compared with birds that forage on other types of food [34], potentially because this facilitates optimal foraging for the caloric content of nectar and ripe fruit.

Taste preferences can also be maladaptive, however, if they lead to the preferential consumption of human foods. The introduction of human foods in urban areas causes changes in resource availability, including calories and nutrient concentration [44,45], as much of the human foods accessible to animals consists of discarded 'junk food' and snacks that are high in fat, sugar and salt. The availability of these foods can alter natural nutritional landscapes with detrimental effects [24,46]. For example, hibernation in mammals is perturbed when human foods are consumed [47,48], and racoons feeding on human foods in urbanized areas have greater weight and blood sugar content [49]. In birds, American crow (*Corvus brachyrhynchos*) chicks are smaller [50], Canadian geese (*Branta canadensis maxima*) have higher rates of angel wing disorder in urbanized areas, probably as a result of nutrient deficiency [51], and Australian magpies (*Gymnorhina tibicen*) show alterations to blood chemistry following backyard provisioning [52]. If animals are adapted to detect and prefer foods based on taste profiles that are coincidentally elevated in human foods, then they could assess these foods erroneously as high quality and favour consumption [24,45]. On the other hand, if their taste receptors are adapted to prefer flavours characteristic of human foods, then they might be better able to take advantage of this newly available resource if native foods decline (i.e. poor condition is preferable to starvation). Therefore, preferential consumption of different foods due to taste preferences, especially in the context of urbanization, could be either adaptive or maladaptive. However, it remains largely unknown whether taste preferences actually underlie the preferential consumption of human foods in urban areas.

Here, we investigate if Darwin's finches, which on human-inhabited islands are known to vary in their preferential consumption of human foods [14], have taste preferences for flavours associated with human foods. Darwin's finches are a model system for demonstrating how foraging ecology shapes adaptation into different ecological niches and results in an adaptive radiation [53,54]. Contemporary work has shown that the presence of humans is affecting traits such as beak morphology [55], and this has been linked to the consumption of human foods [21]. We know that Darwin's finches on human-inhabited islands can preferentially consume human foods including crisps (potato chips), biscuits (hard cookies) and rice at sites where these foods are abundant such as at tourist beaches and in urban areas [14]. Indeed, urban finches can have higher nesting success than non-urban finches [56], suggesting that at least some species of Darwin's finches can become locally

adapted to urban environments. Thus, the Darwin's finches on the Galapagos represent an excellent opportunity to test whether taste preferences might underpin consumption of human foods where these are available and abundant, such as in urban or tourist areas.

We assessed taste preferences in Darwin's finches in separate populations with little to no movement of finches among sites [57] that varied in the availability of human foods previously shown to be attractive to finches [14]. We presented flavours in the absence of visual cues typical to human foods to exclude the role of learned associations with packaging (e.g. crisp packets [14]). If taste preferences arise because of the consumption of human foods, we predicted finches at sites with human foods would only prefer flavours associated with human foods (e.g. sweet, salty and oily) at sites where human foods are abundant. If taste preferences are innate (i.e. latent), we predicted preferences would occur for such flavours across sites, regardless of the presence or absence of human foods. Potential taste preferences can be quantified as variation in feeding rate, or as variation in post-feeding beak-wiping rate [30,58–60]. Although beak-wiping can clean the beak of debris [61], vigorous beak-wiping commonly occurs after feeding on something unpalatable [30,60]. Therefore, we measured taste preferences in terms of consumption and beak-wiping behaviour compared with two controls, neutral pastry that had no flavour added and bitter-flavoured pastry as a negative control. Bitter substances are aversive to many bird species, so we expected finches to consume less bitter-flavoured pastry compared with neutral or other flavoured pastry across all sites as well as to elicit more beak-wiping behaviour [30,60].

# 2. Material and methods

## 2.1. Study species and location

We focused on Darwin's finches, an endemic group of passerines on the Galapagos Islands, at three sites on Santa Cruz Island that varied in their exposure to human foods (electronic supplementary material, figure S1). The remote site was a non-urban site 12 km from the main urban town with no presence of human foods (El Garrapatero proper) [27]; the beach site was El Garrapatero beach, a tourist, non-urban site 12 km from Puerta Ayora where visitors often bring picnics so human food is present and abundant [14,27], and the town site was Puerto Ayora, a fully urbanized town where humans and their food are ubiquitous throughout the entire town [14]. The two focal species were small ground finches (*Geospiza fuliginosa*) and medium ground finches (*Geospiza fortis*). Galápagos mockingbirds (*Mimus parvulus*) and two other finch species (the cactus finch, *Geospiza scandens* and the small tree finch, *Camarhynchus parvulus*) were also present occasionally, but rarely interacted with our experiments.

## 2.2. Experimental protocol and video recording

We conducted trials at each of the three sites (remote = 16 trials, beach = 15 trials, and town = 18 trials; electronic supplementary material, table S1) using a 'cafeteria' tray experiment [14]. Five plastic cups with the diameter of a large chicken egg were randomly positioned in the periphery of a $3 \times 3$ egg carton and placed on the ground on a white plastic dinner plate that was visible below the egg carton (e.g. the carton did not hang over the sides of the plate; electronic supplementary material, figure S2). The central dimple of the egg carton was weighted with a small rock, and the unused dimples were left empty. The trial began when the first bird approached the tray and fed, and then continued for 10 min [14]. If no finches fed, the trial was aborted after 20 min. All trials were performed during the rainy season from 20 February to 25 March 2018, between 6.00 and 11.00 am, or 15.00 and 18.00 and were filmed using a video camera (Sony HDR-CX625 Full HD Compact Camcorder or Canon 7D Mark II with 100–400 mm lens) positioned 10 m from the cafeteria tray. The majority of individuals were not uniquely identifiable (fewer than 4%), so we cannot be sure that birds participating in different trials were independent. However, to reduce the potential for pseudo-replication between trials, we conducted each trial at least 100 m apart within the study locations.

Each cup was filled with 2.5 g of pastry made from flour, unsalted butter and water, following methods from Speed *et al.* [62]. The pastry (335 g flour, 135 g unsalted butter and 30 g water) was flavoured according to commonly available human foods in the environment [14] and each cup was coloured (blue, green, pink, purple and yellow) to facilitate recognition of the contents: (i) blue indicated high in fat (6 g vegetable oil/pastry batch), (ii) green indicated bitter (0.1 g quinine/pastry batch), (iii) purple indicated sweet (23 g sugar/pastry batch), (iv) yellow indicated salty (1.333 g salt/pastry batch) and (v) pink indicated neutral or unflavoured pastry. To habituate the birds to the

experimental set-up, we first conducted trials at each site with only unflavoured pastry (remote = 17 trials, beach = 17 trials, town = 19 trials; electronic supplementary materials, table S1). Birds can have latent colour preferences, either from experience or evolutionary history (e.g. [63]). However, we detected no strong biases within finch species towards, or against, any of the coloured cups based on these trials (electronic supplementary materials, table S2 and figures S3 and S4). Therefore, any preferences detected using flavoured pastry were most likely due to taste and not visual preferences.

## 2.3. Video analysis

Videos were analysed using BORIS (Behavioural Observation Research Interactive Software) [64] and each 10 min trial was analysed by one observer (DL). Species identification was done based on bill and body size in comparison with the size of the cafeteria tray. The observer was trained by KMG to first identify still images of birds (taken from videos collected during trials) and then by using the videos. We counted the number of feeding events at the level of the trial and assigned these to each species of finch. We defined a feeding event as when a bird's beak was submerged into a cup, lifted, and then food was consumed (electronic supplementary material, figure S2). Following each feeding event, we then recorded the number of times the finch wiped its beak on a surface within 20 s in accordance with published methods on beak-wiping [60,61].

## 2.4. Statistical analyses

Statistical analyses were undertaken using the R environment v. 4.0.2 ([65]; data are available as electronic supplementary material). To analyse differences in taste preferences, we used generalized linear mixed-effect models (GLMMs) with a negative binomial error distribution (using the glmer.nb() function in the lme4 package; [66]) to account for overdispersion in the number of feeding events per trial (response variable). Species nested within the trial was included as a random effect to account for non-independence of feeding events, and the fixed effects were species, site and pastry flavour. The reference level (i.e. the model intercept) was 'neutral flavour (pink)' and 'town'. We included an interaction between site and pastry flavour, and assessed whether it contributed significantly to model fit using a likelihood ratio test (compared with a simpler model with the same random effect structure, but containing only additive fixed effects). We then used $z$-tests to assess the significance of differences in consumption among pastry flavours. We report estimates and standard errors and provide incidence rate or odds ratios (negative binomial and binomial models, respectively) to compare effects.

The number of beak wipes following a feeding event were low (5 or fewer) and highly right-skewed (medium ground finch = 3.69, small ground finch = 3.44; calculated using the 'moments' package; [67]) so we therefore modelled the occurrences of beak wipes using a binomial distribution, where the denominator in the response variable was the number of feeding events when no beak wipes occurred. Assumptions of homogeneity of variance and uniformity of the residuals for all models were checked using Kolmogorov–Smirnov tests for uniformity, simulation tests for dispersion and a binomial test for outliers (implemented using the 'DHARMa' package [68]). The cactus finch and small tree finch rarely came to the experimental trays, so only medium and small ground finches were included in the dataset (see electronic supplementary material, figures S3 and S4).

# 3. Results

A total of 49 taste preference trials were conducted across three sites. Participation was similar for both species (medium ground finch = 16 remote trials and 10 town trials, small ground finch = 10 remote trials and 14 town trials) except for trials conducted at beach sites, where medium ground finches participated in only 4 of the 15 trials, whereas small ground finches participated in all trials (electronic supplementary material, table S1).

## 3.1. Taste preferences

The overall consumption of pastry among ground finch species did not differ (1089 feeding events by medium ground finches versus 1186 feeding events by small ground finches; estimate = −0.102 ± 0.163, $z = -0.625$, $p = 0.532$). However, we found some evidence that ground finches preferred some flavour

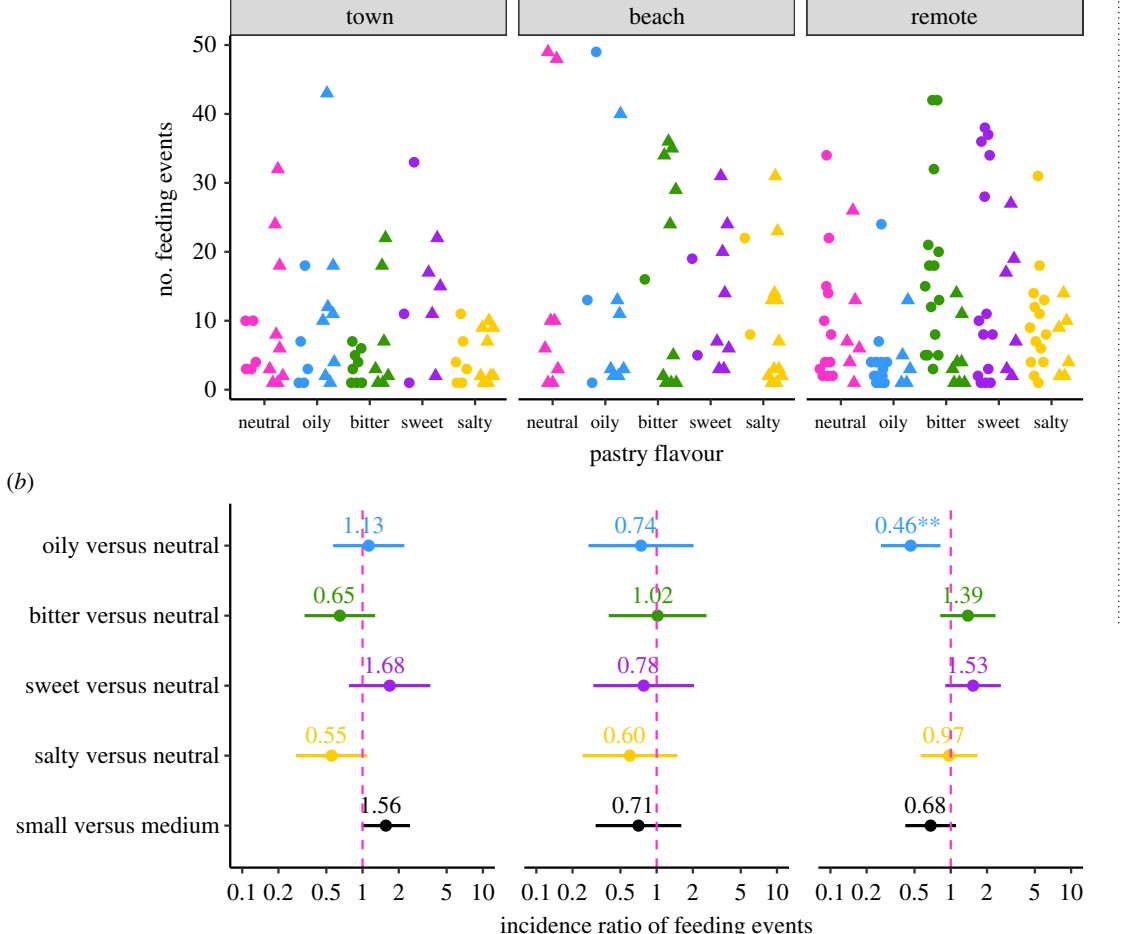

**Figure 1.** Differences in the number of feeding events by medium and small ground finches presented with coloured cups containing pastry with neutral (pink), oily (blue), bitter (green), sweet (purple) or salty (yellow) flavours at either town ($N = 18$ trials), beach ($N = 15$ trials) or remote ($N = 16$ trials) sites; (*a*) presents the raw data; (*b*) presents the effect sizes of the differences between each flavour and the neutral pastry, or between species, computed from GLMMs (see electronic supplementary material, methods for more details). Effects significantly different from zero (dashed pink vertical line) are indicated by asterisks (**0.001 < *p* < 0.01, *0.01 < *p* < 0.05).

types more than others at different sites (flavour type × site, $\chi^2 = 16.352$, d.f. = 8, $p = 0.0376$; figure 1*a,b*; electronic supplementary material, table S3). To investigate this interaction further, we explored differences among flavour types in separate models for each site. At beach sites, there was no significant difference in the number of feeding events according to flavour type ($\chi^2 = 2.025$, d.f. = 4, $p = 0.731$, $N = 15$ trials). However, at remote ($\chi^2 = 20.090$, d.f. = 4, $p = 0.0005$, $N = 16$ trials) and town ($\chi^2 = 10.595$, d.f. = 4, $p = 0.032$, $N = 18$ trials) sites, we detected significant variation in consumption and the preferred flavours differed. Ground finches at remote sites fed less often on oily (blue) pastry than either the neutral or the bitter control whereas at sites in town, finches fed more often on sweet (purple) pastry, but only in comparison with the bitter control (table 1 and figure 1*a,b*).

We next assessed behavioural wiping responses to each flavour type (figure 2). Overall, small ground finches were much more likely to wipe their beak after feeding than medium ground finches (estimate = $0.528 \pm 0.198$, $z = 2.672$, $p = 0.008$; electronic supplementary material, table S4). Again, however, we detected that beak-wiping was more likely to occur after consuming some flavours at some sites (flavour type × site, $\chi^2 = 28.132$, d.f. = 8, $p = 0.0005$; figure 2; electronic supplementary material, table S4). On further inspection (as above), at town sites ground finches showed no difference in their propensity to wipe their beak after feeding on flavoured pastry (table 1, figure 2, $N = 18$ trials). However, at beach sites, ground finches wiped their beaks less often after consuming oily (blue) or

**Table 1.** Mean differences (±s.e.) in the (I) number of feeding events and (II) proportion of feeding events followed by beak-wiping at (a) town sites, (b) beach sites and (c) remote sites that vary in exposure to human foods. Medium ground finches (I: $N = 130$ observations from 26 trials, II: $N = 95$ observations from 26 trials) and small ground finches (I: $N = 195$ observations from 39 trials, II: $N = 114$ observations from 39 trials) were presented with coloured cups containing pastry flavoured to be oily (blue), sweet (purple) or salty (yellow), and two controls: neutral (pink, set as model intercept) and bitter (green). Differences in (I) were estimated using GLMMs with a negative binomial error distribution and where trial and site were included as random effects. Differences in (II) were estimated using a similar model but with a binomial error distribution to account for proportional response data. Significant differences are indicated in italics.

| | I. feeding events | | | II. beak-wiping | | |
|---|---|---|---|---|---|---|
| | mean difference ± s.e. | Z | p | mean difference ± s.e. | Z | p |
| **(a) town sites:** | | | | | | |
| intercept (neutral) | 1.872 ± 0.290 | 6.447 | <0.001 | −2.696 ± 0.345 | −7.489 | <0.001 |
| oily (blue) | 0.118 ± 0.347 | 0.340 | 0.734 | 0.905 ± 0.433 | 2.090 | 0.146 |
| bitter (green) | −0.435 ± 0.345 | −1.262 | 0.207 | −0.053 ± 0.403 | −0.131 | 0.393 |
| sweet (purple) | 0.519 ± 0.397 | 1.310 | 0.190 | 0.936 ± 0.368 | 2.541 | 0.070 |
| salty (yellow) | −0.591 ± 0.348 | −1.699 | 0.089 | 0.593 ± 0.399 | −1.486 | 0.369 |
| species (small) | 0.446 ± 0.236 | 1.888 | 0.059 | 0.565 ± 0.320 | −0.180 | 0.857 |
| **(b) beach sites:** | | | | | | |
| intercept (neutral) | 3.120 ± 0.554 | 5.635 | <0.001 | −1.396 ± 0.429 | −3.255 | 0.001 |
| oily (blue) | −0.300 ± 0.513 | −0.582 | 0.560 | *−0.820 ± 0.414* | *−1.980* | *0.048* |
| bitter (green) | 0.018 ± 0.477 | 0.038 | 0.970 | *−0.733 ± 0.347* | *−2.112* | *0.035* |
| sweet (purple) | −0.249 ± 0.492 | −0.506 | 0.613 | −0.647 ± 0.375 | −1.725 | 0.085 |
| salty (yellow) | −0.512 ± 0.463 | −1.107 | 0.268 | −0.030 ± 0.341 | −0.089 | 0.929 |
| species (small) | −0.348 ± 0.418 | −0.832 | 0.406 | 0.253 ± 0.352 | 0.720 | 0.471 |
| **(c) remote sites:** | | | | | | |
| intercept (neutral) | 2.241 ± 0.240 | 9.328 | <0.001 | −2.969 ± 0.324 | −9.172 | <0.001 |
| oily (blue) | *−0.766 ± 0.291* | *−2.638* | *0.008* | *0.874 ± 0.436* | *2.006* | *0.045* |
| bitter (green) | 0.328 ± 0.270 | 1.216 | 0.224 | 0.210 ± 0.380 | 0.553 | 0.580 |
| sweet (purple) | 0.427 ± 0.271 | 1.575 | 0.115 | *1.314 ± 0.334* | *3.915* | *<0.001* |
| salty (yellow) | −0.033 ± 0.276 | −0.119 | 0.905 | *0.982 ± 0.365* | *2.689* | *0.007* |
| species (small) | −0.385 ± 0.247 | −1.558 | 0.119 | *1.020 ± 0.240* | *4.257* | *<0.001* |

bitter (green) flavoured pastry compared with neutral (pink) pastry and less often after eating salty (yellow) compared with bitter pastry (table 1, figure 2, $N = 15$ trials). At remote sites, ground finches were more likely to wipe their beaks after consuming pastry containing any 'human food' flavour as compared with either the neutral or the bitter control (table 1, figure 2, $N = 16$ trials). Taken together, these results suggest that ground finches in town showed little preference for human food flavours whereas at remote sites, all human food flavours evoked more behavioural reactions than controls, and the oily flavoured pastry was preferred the least.

## 4. Discussion

Taste preferences have often been overlooked in understanding animals' foraging decisions, yet in human-modified environments, latent taste preferences could explain why some species are able to readily adapt to novel foods while others do not. Here, we investigated if taste preferences can explain preferential consumption of human foods by Darwin's finches [14]. We predicted that if finches at sites with more exposure to human foods (i.e. at the tourist beach or in town) showed greater consumption and reduced aversive behavioural responses to flavours typical of these foods

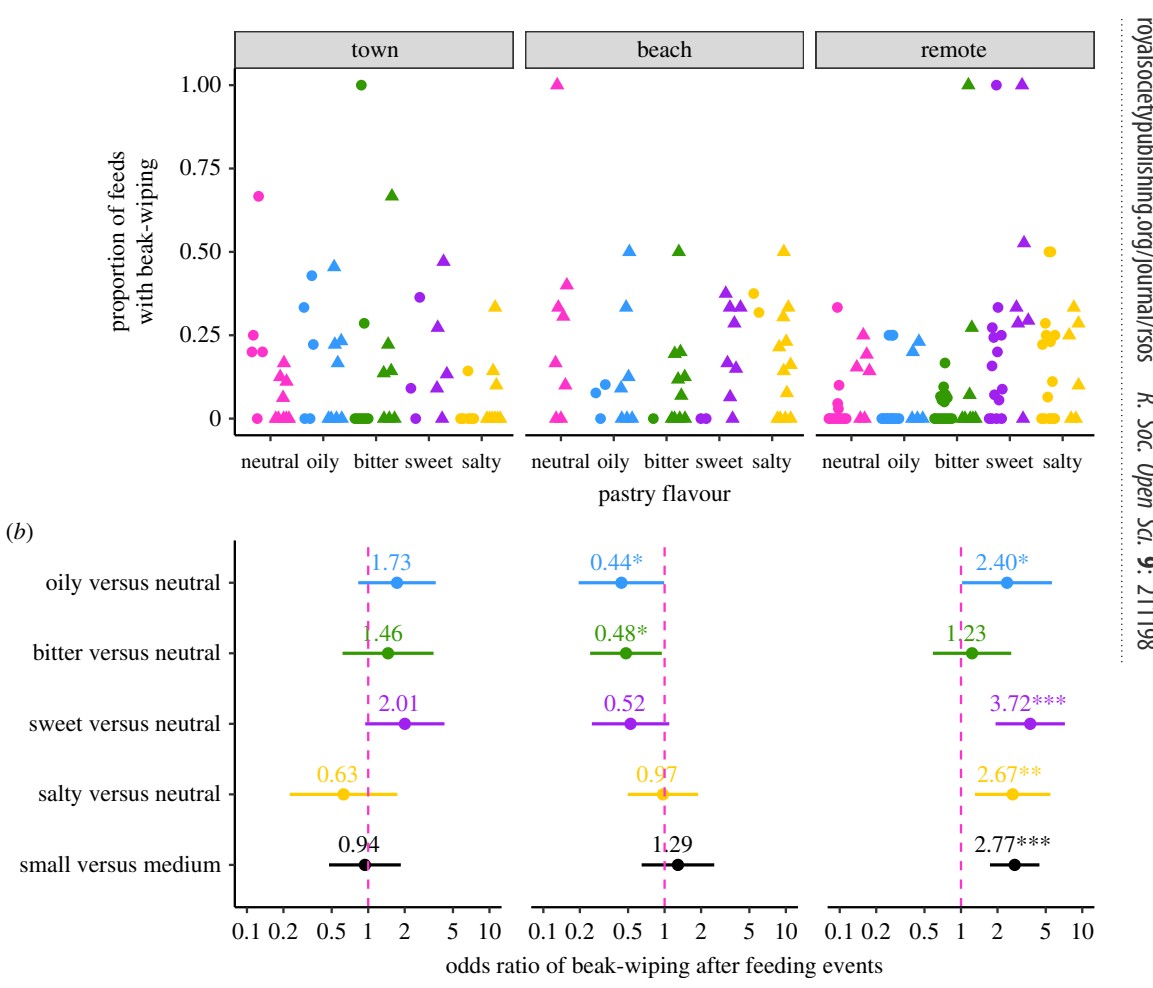

**Figure 2.** Differences in the proportion of feeding events that were followed by beak-wiping by medium and small ground finches presented with coloured cups containing pastry with neutral (pink), oily (blue), bitter (green), sweet (purple) or salty (yellow) flavours at either town ($N = 18$ trials), beach ($N = 15$ trials) or remote ($N = 16$ trials) sites; (a) presents the raw data, (b) presents the effect sizes of the differences between each flavour and the neutral pastry, or between species, computed from GLMMs (see electronic supplementary material, methods for more details). Effects significantly different from zero (dashed pink vertical line) are indicated by asterisks (*** $p < 0.001$, ** $0.001 < p < 0.01$, * $0.01 < p < 0.05$).

(salty, oily and sweet), then taste preferences could have developed from experience with the changed foraging landscape. However, if finches across sites preferred these 'human food flavours' then latent taste preferences could have facilitated rapid adoption of human foods into the diet. Against both predictions, however, the only evidence we detected for a taste preference was that ground finches at remote sites showed a weak aversion to oily foods. This was further supported by behavioural responses, with beak-wiping (an aversive response common across birds [30,60]) occurring more often at remote sites after finches tasted flavours associated with human foods. This suggests that ground finches do not have latent taste preferences for human foods nor have acquired taste preferences from contact with human foods. Furthermore, ground finches at sites with human foods might now be more tolerant of oily flavours and lost their aversion to tastes associated with human foods. It is possible that our sample sizes were too small to detect preferences for human food flavours, or that we did not add sufficient flavour to the pastry for these to be detectable. However, previous work detecting taste preferences had smaller sample sizes than our study (e.g. $n = 6$ per group [36]; $n = 11$ [37]; $n = 6$ and 10 [38]), and the amount of constituents that we added to the pastry emulated human foods as closely as possible. It therefore seems unlikely that our results can only be explained by methodological issues. Since Darwin's finches consume human foods preferentially when available, why did we not find preferences for flavours associated with commonly available human foods?

It could be that Darwin's finches have not evolved a preference for tastes associated with human foods because these species are generalist feeders, especially during periods of non-drought [13]. We conducted our trials during the traditional rainy season when the finches would be more generalist. Taste preferences evolve when they allow animals to identify food items that offer important nutrients (e.g. high in lipids, salts or sugars) [30,31,34,46]. Yet for generalists, it might not be adaptive to have latent taste preferences if these limit individuals from consuming a wide variety of dietary items [13] or they might not need to discern specific foods that are high in lipids, salts or sugars. Indeed, many studies that have found taste preferences in birds have been conducted with specialists [37,38]. Another possibility is that Darwin's finches have not yet acquired preferences for flavours associated with human foods. Tourism has grown exponentially only relatively recently [69]. At the tourist beach site, easy access to the public only became available around 2010 (J. Podos 2020, personal communication), and the town of Puerto Ayora was established in 1926 ([70]; E. Hennessey 2020, personal communication), so perhaps not enough generations have passed from when finches gained access to human foods for finches to acquire taste preferences.

For the finches, we found that the presence or absence of human foods did not correlate with different taste preferences as predicted. Surprisingly, we also found little evidence that the bitter-tasting control elicited beak-wiping. This was unexpected because we know birds possess TAS2R bitter taste receptors [33] and often discriminate against toxic prey via bitterness. In fact, of the species studied by Wang & Zhao [33], medium ground finches had the second most TAS2R genes. So why did we not find a lack of aversion to bitter tastes? If the natural foods found at remote sites are bitter in taste, then perhaps finches are accustomed to bitter tastes. Bitter tastes are often associated with aposematic prey, and the relative lack of aposematic prey on the Galapagos [71] suggests finches may not need to discriminate against bitter tastes. Another possibility is zoopharmacognosy, where animals eat medicinally advantageous foods, despite possible aversive qualities. This is common in birds; great bustards (*Otis tarda*) ingest toxic blister beetles to control digestive tract parasites [72], and house sparrows (*Passer domesticus*) ingest leaves containing quinine (our bittering agent) during malaria outbreaks [73], alleviating symptoms. Quinine is an invasive plant found on the Galapagos. However, no finch has ever been observed consuming quinine (Heinke Jäger 2020, personal communication), so this possibility is unlikely.

The Galapagos Islands are experiencing an exponential increase in urbanization and tourism, including permanent human residents [69], and we know Darwin's finches preferentially consume human foods over natural food sources when readily available [14]. However, here we found little evidence for taste preferences for flavours associated with human foods during the traditional rainy season, although finches at remote sites showed a small aversion towards oily foods. It therefore seems likely that finches do not have latent preferences for these flavours, nor have they acquired a preference through repeated exposure to human foods. Why then have Darwin's finches adapted rapidly to changing food availability and incorporated human foods into their diet when human foods are readily available? One possibility is that they could be attracted to other sensory cues such as aural or visual cues associated with human foods. For example, in town and on the beach (but not in remote areas), finches respond to brightly coloured visual cues of human food packaging and are attracted to the 'crinkle' sound associated with foil and plastic food packaging (electronic supplementary material, figure S5, [14]). Alternatively, it could be driven by availability itself at the beach and town sites. While the food sources Darwin's finches normally feed upon are available within town and at the beach [14] (though we did not control for this), the abundance of human foods at these sites simply make these types of food more accessible to finches, and therefore, finches did not need to discriminate between different flavours typically associated with human foods and aversive flavours to expand their diet diversity [14]. Further work is required to understand the mechanisms underlying how Darwin's finches developed a preference for consuming human foods at sites where human foods are readily available.

Humans, through processes such as urbanization, can have a major impact on foraging ecology by introducing novel foods that can become preferentially consumed by birds [16,17]. However, the mechanisms leading to changes in foraging ecology remain largely unknown. Although we cannot yet explain *why* Darwin's finches prefer human foods, our results help to rule out the possibility that taste preferences play an important role in incorporating human foods into their diets. Similarly, our finding that ground finches do not find bitter tastes aversive expands the increasing knowledge on variation in response to tastes among species. As the adoption of human foods into animals' diets can have cascading effects on health, reproduction and fitness [6,15,20,47,49], it remains of paramount importance to elucidate why some species integrate these foods while others do not.

Ethics. This research was approved by the Galápagos National Park with permit PC-03-18.

Data accessibility. Data are accessible as electronic supplementary material and are archived on Dryad: https://doi.org/10.5061/dryad.dncjsxm0h.

Authors' Contributions. D.L.: data curation, formal analysis, investigation, methodology, writing—original draft and writing—review and editing; L.V.R.: data curation, investigation, writing—original draft and writing—review and editing; R.T.: conceptualization, data curation, formal analysis, funding acquisition, investigation, methodology, resources, supervision, validation, visualization, writing—original draft and writing—review and editing; K.M.G.: conceptualization, data curation, formal analysis, funding acquisition, investigation, methodology, project administration, resources, supervision, validation, writing—original draft and writing—review and editing. All authors gave final approval for publication and agreed to be held accountable for the work performed therein.

Competing interests. We declare we have no competing interests.

Funding. During this study, R.T. was supported by an Independent Research Fellowship from the Natural Environment Research Council UK (NE/K00929X/1) and a start-up grant from the Helsinki Institute of Life Science (HiLIFE), University of Helsinki. K.M.G. was funded by Christ's College and Clare Hall at the University of Cambridge and was supported by Banting Postdoctoral Fellowship from the Natural Sciences and Engineering Research Council of Canada.

Acknowledgements. We thank Lotte Skovmand for their assistance with fieldwork and Nick Davies for their assistance with the research project.

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
