## [Peer Review File · Royal Society Open Science]

Review History

RSOS-211198.R0 (Original submission)

Review form: Reviewer 1

Is the manuscript scientifically sound in its present form?

Yes

Are the interpretations and conclusions justified by the results?

Yes

Is the language acceptable?

Yes

Do you have any ethical concerns with this paper?

No

Have you any concerns about statistical analyses in this paper?

No

Recommendation?

Accept with minor revision (please list in comments)

Comments to the Author(s)

This study represents an interesting addition to the growing literature on the behavioural responses of fauna to urbanisation. The experimental setup and analyses are adequate, and the presentation of results is clear. The discussion is appropriate. I have made a few suggestions in the manuscript (see Appendix A) that I hope are useful. I would like to recommend to the authors to review the available literature on scrounging behaviour of Kea in NZ of seagulls in Australia and NZ as these are stark examples of behavioural changes in response to junk food availability. The authors should acknowledge explicitly other limitations of their study, such as seasonality, and lack of control on availability of native foodstuffs. Overall a neat study.

Review form: Reviewer 2**Is the manuscript scientifically sound in its present form?**

Yes

Are the interpretations and conclusions justified by the results?

No

Is the language acceptable?

Yes

Do you have any ethical concerns with this paper?

No

Have you any concerns about statistical analyses in this paper?

Yes

Recommendation?

Major revision is needed (please make suggestions in comments)

Comments to the Author(s)

I have a problem with the whole manuscript. The authors focused on really important and interesting problem of food taste choice, but design really does not provides final solution. I put my comments also on the MS (see Appendix B). I also suggest some changes in statistics.

Decision letter (RSOS-211198.R0)

Dear Dr Gotanda

The Editors assigned to your paper RSOS-211198 "Darwin's finches can have taste preferences, but not for human foods" have now received comments from reviewers and would like you to

revise the paper in accordance with the reviewer comments and any comments from the Editors. Please note this decision does not guarantee eventual acceptance.

Please submit your revised manuscript and required files (see below) no later than 21 days from today's (ie 13-Sep-2021) date. Note: the ScholarOne system will 'lock' if submission of the revision is attempted 21 or more days after the deadline. If you do not think you will be able to meet this deadline please contact the editorial office immediately.

on behalf of Dr Dieter Lukas (Associate Editor) and Kevin Padian (Subject Editor)
openscience@royalsociety.org

Associate Editor Comments to Author (Dr Dieter Lukas):

Associate Editor: 1

Comments to the Author:

Dear authors,

Your article entitled "Darwin's finches can have taste preferences, but not for human foods" has now been seen by two reviewers and the reviewers' comments are appended below. As you will see, both reviewers consider your study to contribute relevant insights into how sensory information might relate to avian adaptation to urbanisation, and I share their views. Yet they have several comments that need to be addressed carefully before I would consider recommending this article for publication.

The main issues raised are about the limitations of this study. Studies such as these on wild birds are difficult to plan and conduct, so any additional information gained can be helpful. However, the limitations should be acknowledged and the potential implications of the results tempered accordingly.

I agree with reviewer 2 that the small sample sizes, due to birds often not participating in the experiments, limit the ability to draw firm conclusions about the behaviour under investigation. You acknowledge this limitation in the discussion, but come to the conclusion that it "seems unlikely that our results can only be explained by methodological issues" (line 245f). You base this conclusion on the observation that some studies with small sample sizes did find effects. However, there might well be publication bias (as has been shown for essentially every other research question), with small studies that did not find an effect underreported. With small sample sizes, power is low to detect effect, which could mean that you are in the proportion of cases where you did not find an effect even though it is there. This appears further confounded by the fact that your sample sizes might be even smaller than what you report. For one, you mention that you "cannot be sure that birds participating in different trials were independent", and in addition you also might have had social effects that mean the observations are not fully independent. I would therefore ask you to edit your manuscript to more clearly state your findings in the lights of this limitation. One change would be to add the actual sample sizes to each reported result (after every p-value). The other would be to make less definite statement about the implications of what you found. For example, the final two sentences of the abstract state that you found support for the hypotheses that ground finches "do not find bitter-tasting food aversive" and that "taste preferences are unlikely to play a major role". Phrasing it in terms of saying that you found no evidence that ground finches find bitter-tasting food aversive and that taste preferences are linked to the consumption of human food can make it clearer that these are the conclusions within the limitations of your study rather than a general statement about the species.

In addition, I agree with reviewer 1 that also other limitations besides sample size, such as seasonality or lack of control on availability of native foodstuffs, should be more clearly acknowledged. This again limits the power you have to draw definite conclusions, also supporting the changes to the statements of the implications of your study.

In addition to these main issues, both reviewers also have provided annotations on your manuscript with further more specific comments on particular sentences.

All the comments focus on changing the presentation of results, changing the phrasing in places and adding information to the manuscript that either presumably was already part of your decision processes or can be added through further further references. Addressing the comments appears feasible, and I am looking forward to read a new improved version of the manuscript.

Reviewer comments to Author:

Reviewer: 1

Comments to the Author(s) - see also attachment 'Colour and Taste Preferences Lever Manuscript'.

This study represents an interesting addition to the growing literature on the behavioural responses of fauna to urbanisation. The experimental setup and analyses are adequate, and the presentation of results is clear. The discussion is appropriate. I have made a few suggestions in the manuscript (see Word document attached) that I hope are useful. I would like to recommend to the authors to review the available literature on scrounging behaviour of Kea in NZ of seagulls in Australia and NZ as these are stark examples of behavioural changes in response to junk food availability. The authors should acknowledge explicitly other limitations of their study, such as seasonality, and lack of control on availability of native foodstuffs. Overall a neat study.

Reviewer: 2

Comments to the Author(s) - see also attachment 'RSOS-211198_Proof_hi'.

I have a problem with the whole manuscript. The authors focused on really important and interesting problem of food taste choice, but design really does not provides final solution. I put my comments also on the MS. I also suggest some changes in statistics.

===PREPARING YOUR MANUSCRIPT===

===PREPARING YOUR REVISION IN SCHOLARONE===

Author's Response to Decision Letter for (RSOS-211198.R0)

See Appendix C.

Decision letter (RSOS-211198.R1)

Dear Dr Gotanda

The Editors assigned to your paper RSOS-211198.R1 "Darwin's small and medium ground finches might have taste preferences, but not for human foods" would like you to revise the paper in accordance with the comments below. Please note this decision does not guarantee eventual acceptance.

We invite you to respond to the comments supplied below and revise your manuscript. Below the Editor's comments we provide additional requirements. Final acceptance of your manuscript is dependent on these requirements being met. We provide guidance below to help you prepare your revision.

Please submit your revised manuscript and required files (see below) no later than 21 days from today's (ie 08-Nov-2021) date. Note: the ScholarOne system will 'lock' if submission of the revision is attempted 21 or more days after the deadline. If you do not think you will be able to meet this deadline please contact the editorial office immediately.

on behalf of Dr Dieter Lukas (Associate Editor) and Kevin Padian (Subject Editor)
openscience@royalsociety.org

Associate Editor Comments to Author (Dr Dieter Lukas):
Associate Editor
Comments to the Author:
Dear authors,

Thank you for replying to the reviewers' and my comments in such detail, and for changing the phrasing in your manuscript to reflect the potential limitations.

I have a follow-up to the critical comments reviewer 2 made. I realise that their initial comments were not very detailed, and I also had not been entirely sure what they were criticising. After reading your replies, I now am wondering the following about their comments about the setup and the stimuli involved.

If I understand your setup correctly, each trial was performed at a different location. Accordingly, you assume that for each bird that participates, they only see the setup once when their data is being collected. That would mean that birds coming in are naive to the tastes, and in particular to the fact that a given colour indicates a particular flavour. Doesn't that mean that for the analyses you have to exclude all zeroes? A bird not feeding from a particular cup does not tell you

anything about the preference or aversion for that respective taste because that bird does not know that the taste is there. These birds cannot be said that they are avoiding that cup because of the taste, because they never experienced the taste of the pastry in that cup. So I think the only meaningful information is in what happens after a bird actually starts feeding from a given cup: do they continue or do they stop because they now have experienced that taste. That means you see whether the feeding rate of those events where a bird started feeding is higher for some flavours than for others. I think this has to be at the level of individual birds, rather than the non-zero feeding rate during a given trial. In addition, how would a bird know that the pastry in a cup different from the one they started with at random would taste different? So why would you expect a bird that has started on a neutral cup to stop feeding there and switch to for example the cup with the sweet pastry? If birds are truly naive, wouldn't you expect that after a bird picks one of the cups at random it just continues eating there - which would mean you would not expect to be able to tell whether there are taste differences with this setup? If I am wrong, please add an explanation to the manuscript why you are predicting different feeding rates for the different flavours with your setup.

Two minor comments:

I am not familiar with these species, so it was sometimes confusing to read about medium and small ground finches because I expected this to potentially reflect multiple species that are grouped by size, rather than each term referring to one specific species. Maybe you could already in the abstract list the Latin species name when you first introduce medium and small ground finches to clarify this (line 14). Somewhat linked to that, the legend for Figure 1 in line 543 has the "Medium" capitalised whereas in the remainder of the text it is spelled in small letters.

I am not sure why you removed the sample size information from the result section (lines 210, 211, 229, 234). As I mentioned before, I think it would be helpful to have the mention of the sample size throughout. Here especially, because before you only speak about trials whereas these analyses are actually on the level of feeding and beak-wiping events.

===PREPARING YOUR MANUSCRIPT===

If you have been asked to revise the written English in your submission as a condition of publication, you must do so, and you are expected to provide evidence that you have received language editing support. The journal would prefer that you use a professional language editing service and provide a certificate of editing, but a signed letter from a colleague who is a fluent speaker of English is acceptable. Note the journal has arranged a number of discounts for authors using professional language editing services (<https://royalsociety.org/journals/authors/benefits/language-editing/>).

===PREPARING YOUR REVISION IN SCHOLARONE===

Author's Response to Decision Letter for (RSOS-211198.R1)

See Appendix D.

Decision letter (RSOS-211198.R2)

Dear Dr Gotanda

On behalf of the Editors, we are pleased to inform you that your Manuscript RSOS-211198.R2 "Darwin's small and medium ground finches might have taste preferences, but not for human foods" has been accepted for publication in Royal Society Open Science subject to minor revision in accordance with the referees' reports. Please find the referees' comments along with any feedback from the Editors below my signature.

Please submit your revised manuscript and required files (see below) no later than 7 days from today's (ie 14-Dec-2021) date. Note: the ScholarOne system will 'lock' if submission of the revision is attempted 7 or more days after the deadline. If you do not think you will be able to meet this deadline please contact the editorial office immediately.

on behalf of Dr Dieter Lukas (Associate Editor) and Kevin Padian (Subject Editor)
 openscience@royalsociety.org

Associate Editor Comments to Author (Dr Dieter Lukas):

Dear authors,

Thank you for thinking the issues around the design through, and for adjusting the analyses accordingly. I think there are now some insightful findings about the behaviour of these finches around human food flavours!

I only have three remaining comments regarding your implementation of the revisions:

1) Abstract, line 18ff: "Our results suggest, therefore, that while Darwin's finches may have acquired a tolerance to the flavours of human food"

I wonder whether you want to specify that the "Darwin's finches regularly exposed to human food might have acquired a tolerance to those flavours". You did detect a difference between the town compared to the remote site.

2) Discussion, line 302ff: "This suggests that ground finches do not have latent taste preferences for human foods, or have acquired preferences from contact with human foods" The second part, that birds in the site with more human food have no acquired differences, seems to go against your statement in the results ("at remote sites, all human-food flavours evoked more behavioural reactions than controls") and the sentence quoted above from the abstract. I can see that there are no preferences, but it seems like it would be good to clarify that birds at the town site seem to have lost their aversion.

3) The top panels of all the figures still show the full data, including the zeroes. I think that is fine, but it probably would be worth explaining in the methods that you excluded the zeroes and provide explanations for this approach from our exchange; and worth mentioning in the figure legend that they show the full dataset including events where birds did not feed.

===PREPARING YOUR MANUSCRIPT===

one version should clearly identify all the changes that have been made (for instance, in coloured highlight, in bold text, or tracked changes);

===PREPARING YOUR REVISION IN SCHOLARONE===

- Ensure that your data access statement meets the requirements at <https://royalsociety.org/journals/authors/author-guidelines/#data>. You should ensure that you cite the dataset in your reference list. If you have deposited data etc in the Dryad repository, please only include the 'For publication' link at this stage. You should remove the 'For review' link.
- If you are requesting an article processing charge waiver, you must select the relevant waiver option (if requesting a discretionary waiver, the form should have been uploaded, see 'File upload' above).
- If you have uploaded any electronic supplementary (ESM) files, please ensure you follow the guidance at <https://royalsociety.org/journals/authors/author-guidelines/#supplementary-material> to include a suitable title and informative caption. An example of appropriate titling and captioning may be found at https://figshare.com/articles/Table_S2_from_Is_there_a_trade-off_between_peak_performance_and_performance_breadth_across_temperatures_for_aerobic_scope_in_teleost_fishes_/3843624.

Author's Response to Decision Letter for (RSOS-211198.R2)

See Appendix E.

Decision letter (RSOS-211198.R3)

Dear Dr Gotanda,

I am pleased to inform you that your manuscript entitled "Darwin's small and medium ground finches might have taste preferences, but not for human foods" is now accepted for publication in Royal Society Open Science.

on behalf of Dr Dieter Lukas (Associate Editor) and Kevin Padian (Subject Editor)
openscience@royalsociety.org

Appendix A

Darwin's finches can have taste preferences, but not for human foods

Lever, D.¹, Rush, L. V.², Thorogood, R.^{1,3,4,7}, Gotanda, K. M.^{1,5,6,8*}

¹ Department of Zoology, University of Cambridge, Downing Street, Cambridge CB2 3EJ
United Kingdom

² Department of Geology, Laurentian University, 935 Ramsey Lake Rd, Sudbury, Ontario, P3E
2C6, Canada

³ Helsinki Institute of Life Science (HiLIFE), University of Helsinki, Helsinki 00014, Finland.

⁴ Research Program in Organismal and Evolutionary Biology, Faculty of Biological and
Environmental Sciences, University of Helsinki, Helsinki 00014, Finland.

⁵ Département de Biologie, Université de Sherbrooke, 2500, boul de l'Université, Sherbrooke,
Québec, J1K 2R1, Canada

⁶ Department of Biological Sciences, Brock University, 1812 Sir Isaac Brock Way, St.
Catharine's, Ontario, L2S 3A1, Canada

⁷ ORCID: 0000-0001-5010-2177

⁸ ORCID: 0000-0002-3666-0700

* To whom correspondence should be addressed: kg419@cam.ac.uk

Abstract

Urbanization is rapidly changing ecological niches. On the Galapagos Islands, Darwin's finches consume human-introduced foods preferentially; however, it remains unclear why. Here we presented pastry with flavour profiles typical of human foods (oily, salty, sweet) to small and medium ground finches to test if latent taste preferences might drive selection of human foods. If human-food flavours were consumed more than a neutral or bitter control at sites with human foods, then we predicted tastes were acquired after experience with human foods; however, if no site-differences were found then this would indicate latent taste preferences. Contrary to both predictions, we found no evidence that human-food flavours were preferred compared to control flavours. Instead, medium ground finches consumed the bitter control pastry most and wiped their beaks more frequently after feeding on oily and sweet pastry (post-ingestion beak wiping can indicate aversions). Small ground finches showed no differences in consumption but wiped their beaks most after feeding on sweet pastry. Our results suggest that unlike many species, medium and small ground finches do not find bitter-tasting food aversive. Furthermore, taste preferences are unlikely to play a major role in Darwin's finches adaptation to the presence of human foods during increased urbanization.

Key words: *Geospiza fortis*, *Geospiza fuliginosis*, foraging, human influences, urbanization, Galapagos

Introduction

Human behaviour is now recognised to be a strong driver of local adaptation and differences among populations of animals [1,2]. Urbanization [3–5], for example, can have profound effects on foraging because humans often introduce novel foods to the surrounding environment either intentionally (e.g. via garden bird feeding [6,7]) or unintentionally (e.g. by planting ornamental plants [8,9]). This changes the diversity and availability of food items, and generates different foraging landscapes from those in which most animals evolved [10–12]. However, organism's responses to these altered niches vary, with some birds, for example, not only adapting more readily to incorporate human-foods into their diet, but even preferentially consuming them over more natural food sources [11,13–15]. These differences in if and how populations and species utilise human foods can then have consequences for local adaptation [7,16], potentially affect species divergence [17], or mediate the potential for invasive species to establish [18,19]. Furthermore, the gut microbiome, which is increasingly recognised 
[revised manuscript text omitted]
 20 minutes. All trials were performed from February 20th to March 25th, 2018, between 6am and 11am, or 3pm and 6pm and were filmed using a video camera (Sony HDR-CX625 Full HD Compact Camcorder or Canon 7D Mark II with 100-400mm lens) positioned 10 metres from the cafeteria plate. The majority of individuals were not uniquely identifiable, so we cannot be sure that birds participating in different trials were independent. To reduce the potential for pseudo-replication between trials, we moved the location of the cafeteria experiment for each trial by at least 100 m within the study locations.

Each cup was filled with 2.5 g of pastry made from flour, unsalted butter, and water, following methods from Speed and colleagues (2000). The pastry was flavoured according to commonly available human-foods in the environment [11] and each cup was coloured (blue, green, pink, purple, and yellow) to facilitate recognition of the contents: (i) blue indicated high in fat (6g vegetable oil/pastry batch), (ii) green indicated bitter (0.1g quinine/pastry batch), (iii) purple indicated sweet (23g sugar/pastry batch), (iv) yellow indicated salty (1.333g salt/pastry

batch), and (v) pink indicated neutral or unflavoured pastry. To habituate the birds to the experimental set-up, we first conducted trials at each site with only unflavoured pastry (remote = 17 trials, beach = 17 trials, town = 19 trials; Supplemental Materials, Supplemental Table 1). Birds can have latent colour preferences, either from experience or evolutionary history. However, we detected no strong biases within finch species towards, or against, any of the coloured cups based on these trials (Supplemental Materials; Supplemental Table 2; Supplemental Figures 2 & 3).

Video analysis

Videos were analysed using BORIS (Behavioral Observation Research Interactive Software) (Friard and Gamba, 2016) and each 10-minute trial was analysed by one observer (DL). As most birds (fewer than 4%) were not individually identifiable, we counted the number of feeding events at the level of the trial and assigned these to each species of finch based on their body size and beak morphology. We defined a feeding event as when a bird's beak was submerged into a cup, lifted, and then food was consumed. If at least one cup was visited during a trial, any cups not visited received a score of 0. Following each feeding event, we then recorded the number of times the finch wiped its beak on a surface within 20 seconds in accordance with published methods on beak wiping [54,55].

Statistical analyses

Statistical analyses were undertaken using the R environment version 4.0.2 (R Core team, 2020; analyses code and data are available as Supplemental Materials). To analyse differences in taste preferences, we used generalised linear mixed-effect models (GLMMs) with a negative binomial

error distribution (using the `glmer.nb()` function in the `lme4` package; Bates et al. 2015) to account for overdispersion in the number of feeding events (response variable). Trial number was included as a random effect to account for non-independence of feeding events within trials and the fixed effects were site, pastry flavour, and their interaction. If the interaction did not contribute significantly to model fit (using a likelihood ratio test to compare with a simpler model containing only additive fixed effects) it was removed, but site and pastry flavour were retained in all models. We then used z-tests to assess the significance of differences in consumption among pastry flavour and site using ‘neutral flavour (pink)’ and ‘town’ as the reference level (i.e. intercept). We report estimates and standard errors and provide incidence rate or odds ratios to compare effects.

The number of beak wipes following a feeding event were low (5 or fewer) and highly right-skewed (medium ground finch = 3.69, small ground finch = 3.44; calculated using the ‘moments’ package; Komsta and Novomestky 2015) so we therefore we modelled the occurrences of beak wipes using a binomial distribution, where the denominator in the response variable was the number of feeding events when no beak wipes occurred. Assumptions of homogeneity of variance and uniformity of the residuals for all models were checked using Kolmogorov-Smirnov tests for uniformity, simulation tests for dispersion, and a binomial test for outliers (implemented using the ‘DHARMA’ package [58]). The cactus finch and small tree finch rarely came to the experimental trays so only medium and small ground finches were included in the data set. Furthermore, attendance of medium ground finches at beach trials was very low and they did not feed from all cups in any trial. Therefore, data from this location were excluded for medium ground finches and the two species were analysed separately.

Results

A total of 53 taste preference trials were conducted across three sites. Both medium and small ground finches varied in their propensity to participate in experimental trials at different locations. Medium ground finches participated at 16 of the 16 trials conducted at remote sites, 0 of the 15 trials conducted at beach sites, and in 10 of the 18 trials conducted at town sites. Small ground finches participated at 10 of the 16 trials conducted at remote sites, 15 of the 15 trials conducted at beach sites, and 14 of the 18 trials conducted at town sites (Supplemental Table 1).

Medium ground finches

Medium ground finches engaged in significantly more feeding events at the remote site than the town site (Table 1, Figure 1C, Supplemental Table 1b), but we found no evidence that taste preferences differed according to location (location*flavour-type: $\chi^2 = 4.687$, d.f. = 4, $p = 0.321$). Medium ground finches did, however, differ in their feeding events with respect to flavour-type (flavour-type: $\chi^2 = 11.810$, d.f. = 4, $p = 0.019$), with significantly more feeds occurring on the bitter (green) pastry than the neutral flavoured pastry (Table 1; Figure 1A & 1C). No other flavour-types were consumed more frequently than the neutral pastry (Table 1; Figure 1A & 1C). We next assessed behavioural wiping responses to each flavour-type (Figure 1B & 1D). However, as there were fewer than 10 trials at town sites where each flavour was consumed (neutral (pink): $N = 5$, oily (blue): $N = 5$, bitter (green): $N = 8$, sweet (purple): $N = 3$, salty (yellow): $N = 6$); we could not test whether responses to specific flavour-types differed among the locations. Overall, medium ground finches were more likely to wipe their beaks after consuming oily (blue; 14.7% of feeding events) or sweet (purple; 15.5% of feeding events) flavours than after consuming neutral flavoured pastry (pink; Table 1; Figure 1B & 1D) and

wiped their beaks the least after consuming bitter (green; 5% less than after consuming neutral pastry (Figure 1D)). Beak wiping also tended to occur more often after feeding events at town sites compared to remote sites, although this was not significant (Table 1, Figure 1B & 1D). Together, these results suggest that medium ground finches consumed bitter-flavoured pastry the most (at both town and remote sites) yet wiped their beaks the least after feeding on this flavour.

Small ground finches

Small ground finches, on the other hand, showed no significant differences in their feeding preferences (Figure 2), either among flavour-types ($\chi^2 = 1.160$, d.f. = 4, $p = 0.885$, Table 1; Figure 2A & 2C), among the three sites ($\chi^2 = 2.866$, d.f. = 2, $p = 0.239$; town vs. beach estimate = 0.424 ± 0.319 , $z = 1.329$, $p = 0.184$; town vs. remote estimate = -0.112 ± 0.357 , $z = -0.315$, $p = 0.753$; Table 1; Figure 2A & 2C), or in interaction ($\chi^2 = 4.728$, d.f. = 8, $p = 0.786$; Table 1; Figure 2A & 2C). Overall, while there were no differences among locations in the proportion of feeding events that led to beak wiping ($\chi^2 = 2.298$, d.f. = 2, $p = 0.317$; Table 1; Figure 2B & 2D), small ground finches were 63% more likely to wipe their beaks after consuming sweet-flavoured (purple) rather than neutral pastry (pink; Figure 2D). There were no significant differences between other flavours and neutral-flavoured pastry (Table 1; Figure 2B & 2D). The number of feeding trials where each flavour was consumed were too few to statistically test for location-specific differences in beak-wiping responses to flavours. Nevertheless, together the feeding events and beak wiping data suggest that small ground finches show little discrimination of foods based on human-food flavours, regardless of location.

Discussion

Taste preferences have often been overlooked in understanding animals' foraging decisions, yet in human-modified environments, latent taste preferences could explain why some species are able to readily adapt to novel foods while others do not. Here we investigated if taste preferences can explain preferential consumption of human foods by Darwin's finches [11]. We predicted that if finches at sites with more exposure to human foods (i.e. at the tourist beach or in town) showed greater consumption and reduced aversive behavioural responses to flavours typical of these foods (salty, oily, sweet), then taste preferences could have developed from experience with the changed foraging landscape. However, if finches across sites preferred these 'human-food flavours' then latent taste preferences could have facilitated rapid adoption of human foods into the diet. Against both predictions, the only evidence we detected for a taste preference was that medium ground finches favoured the bitter-flavoured control. Medium ground finches fed most often on the bitter-flavoured pastry and wiped their beaks the least following its ingestion. This was surprising, given that bitter-tasting foods often elicit increased beak wiping in birds [25,54]. It is possible that our sample sizes were too small to detect preferences for human-food flavours, or that we did not add sufficient flavour to the pastry to be detectable. However, previous work detecting taste preferences had smaller sample sizes than our study ([e.g. 31 (n = 6/group), 32 (n = 11), 33 (n = 6 and 10)], and the amount of flavour we added to the pastry emulated human foods as closely as possible. It therefore seems unlikely that our results can only be explained by methodological issues. Since Darwin's finches consume human foods preferentially when available, why did we not find preferences for flavours associated with commonly available human foods?

It could be that Darwin's finches have not evolved a preference for tastes associated with human foods because these species are generalist feeders, especially during periods of non-drought [10]. Taste preferences evolve when they allow animals to identify food items that offer important nutrients (e.g., high in lipids, salts, or sugars)[25,26,29,40]. Yet for generalists, it might not be adaptive to have latent taste preferences if these limit individuals from consuming a wide variety of dietary items [10] or they might not need to discern specific foods that are high in lipids, salts, or sugars. Indeed, many studies that have found taste preferences in birds have been conducted with specialists [32,33]. Another possibility is that Darwin's finches have not yet acquired preferences for flavours associated with human foods. At the tourist beach site, easy access to the public only became available around 2010 (J. Podos, personal communication), and the town of Puerto Ayora was established in 1926 [59; E. Hennessey, personal communication], so perhaps not enough generations have passed from when finches gained access to human foods for finches to acquire taste preferences.

For the medium ground finch, we found that the bitter taste is preferred across sites (Figure 1; Table 1), suggesting a latent taste preference for bitter, and the presence or absence of human foods did not correlate with different taste preferences as predicted. We also found a lack of aversion to bitter pastry in both species with finches not wiping their bills more after consuming bitter pastry. This was unexpected because we know birds possess TAS2R bitter taste receptors [28], and often discriminate against toxic prey via bitterness. In fact, of the species studied by Wang and Zhao [28], medium ground finches had the second most TAS2R genes. So why would medium ground finches prefer bitter tastes as opposed to being averse? If the natural foods found at remote sites are bitter in taste, that might explain why medium ground finches

have a latent preference for bitter tastes. Bitter tastes are often associated with aposematic prey, and the relative lack of aposematic prey on the Galapagos [60] suggests finches do not need to develop an aversion to bitter tastes as shown by the preferential consumption of bitter tastes and not wiping their bills after consuming bitter pastry. Another possibility is zoopharmacognosy, where animals eat medicinally advantageous foods, despite possible aversive qualities. This is common in birds; great bustards (*Otis tarda*) ingest toxic blister beetles to control digestive tract parasites [61], and house sparrows (*Passer domesticus*) ingest leaves containing quinine (our bittering agent) during malaria outbreaks [62], alleviating symptoms. Quinine is an invasive plant found on the Galapagos. However, no finch has ever been observed consuming quinine (Heinke Jäger, personal communication), so this possibility is unlikely.

The Galapagos Islands are experiencing an exponential increase in urbanization and tourism, including permanent human residents [63], and we know Darwin's finches preferentially consume human foods over natural food sources when readily available [11]. However, here we found no taste preferences for flavours associated with human foods. It therefore seems likely that finches do not have latent preferences for these flavours, nor acquired a preference through repeated exposure to human foods. Why then have Darwin's finches adapted rapidly to changing food availability and incorporated human foods into their diet? One possibility is that they could be attracted to other sensory cues such as aural or visual cues associated with human foods. For example, in town and on the beach (but not in remote areas), finches respond to brightly coloured visual cues of human food packaging and are attracted to the 'crinkle' sound associated with foil and plastic food packaging [Supplemental Figure 4; 11]. Alternatively, it could be driven by availability itself at the beach and town sites. While the food

sources Darwin's finches normally feed upon are available within town and at the beach [11], the abundance of human foods at these sites simply make these more accessible to finches, and therefore, finches did not need to discriminate between different flavours typically associated with human foods and aversive flavours to expand their diet diversity [11]. Further work is required to understand the mechanisms underlying how Darwin's finches developed a preference for consuming human foods at sites where human foods are readily available.

Humans, through processes such as urbanization, can have a major impact on foraging ecology by introducing novel foods that can become preferentially consumed by birds [13,14]. However, the mechanisms leading to changes in foraging ecology remain largely unknown. Although we cannot yet explain *why* Darwin's finches prefer human foods, our results help to rule out the possibility that taste preferences play an important role in incorporating human foods into their diets. Similarly, our finding that ground finches do not find bitter tastes aversive expands the increasing knowledge on variation in response to tastes among species. As the adoption of human foods into animals' diets can have cascading effects on health, reproduction, and fitness [6,12,16,41,43], it remains of paramount importance to elucidate why some species integrate these foods while others do not.

Author contributions

KMG and RT designed the study, KMG and LVR collected the field data, DL analyzed the videos, RT, DL, and KMG analyzed the data, and all authors wrote and edited the manuscript.

[revised manuscript text omitted]
 by (a) medium ground finches and (b) small ground finches from coloured cups containing pastry flavoured to be salty (yellow), sweet (purple), bitter (green), or oily (blue). Differences in (I) were estimated using generalised linear mixed effects models with a negative binomial error distribution and where trial and site were included as random effects. Differences in (II) were estimated using a similar model but with a binomial error distribution to account for proportional response data. ‘Neutral’, unflavoured pastry was set as the intercept (in parentheses) in all models.

	I. Feeding events			II. Beak wiping		
	Mean difference \pm S.E.	Z	p	Mean difference \pm S.E.	Z	p
(a) Medium ground finches:						
(neutral)	0.689 \pm 0.367	1.879	0.060	-2.696 \pm 0.345	-7.489	<0.001
Oily (blue)	-0.442 \pm 0.367	-1.206	0.228	0.905 \pm 0.433	2.090	0.037
Bitter (green)	0.751 \pm 0.355	2.118	0.034	-0.053 \pm 0.403	-0.131	0.896
Sweet (purple)	0.427 \pm 0.353	1.207	0.227	0.936 \pm 0.368	2.541	0.011
Salty (yellow)	0.207 \pm 0.355	0.584	0.559	0.593 \pm 0.399	1.486	0.137
Site - remote	1.075 \pm 0.359	2.993	0.003	0.565 \pm 0.320	1.765	0.078
(b) Small ground finches:						
(neutral)	1.827 \pm 0.348	5.258	<0.001	-1.815 \pm 0.257	-7.050	<0.001
Oily (blue)	-0.326 \pm 0.428	-0.762	0.446	0.060 \pm 0.276	0.217	0.828
Bitter (green)	-0.152 \pm 0.427	-0.356	0.722	-0.024 \pm 0.274	-0.087	0.931
Sweet (purple)	-0.092 \pm 0.425	-0.215	0.830	0.489 \pm 0.242	2.017	0.044
Salty (yellow)	-0.396 \pm 0.427	-0.929	0.353	0.121 \pm 0.271	0.448	0.654
Location	0.424 \pm 0.319	1.329	0.184	0.144 \pm 0.255	0.563	0.573
	-0.112 \pm 0.357	-0.315	0.753	0.438 \pm 0.291	1.508	0.132

Figure Legend

Figure 1. Differences in the (A,C) number of feeding events and (B,D) proportion of feeding events that were followed by beak-wiping by Medium ground finches presented with pastry in coloured cups with neutral (pink), oily (blue), bitter (green), sweet (purple), or salty (yellow) flavours common to human-foods in either town (●, N = 10 trials) or remote (▲, N = 16 trials) locations. A and B present the raw data, C and D present the effect sizes of the differences between each flavour and the neutral pastry, or between town and remote locations, computed from generalised linear mixed effects models (see Methods for more details). Effects that were significantly different from zero (dashed pink vertical line) are indicated by asterisks (* $0.01 < p < 0.05$, ** $0.001 < p < 0.01$).

Figure 2. Differences in the (A,C) number of feeding events and (B,D) proportion of feeding events that were followed by beak-wiping by small ground finches presented with pastry in coloured cups with neutral (pink), oil (blue), bitter (green), sweet (purple), or salty (yellow) flavours common to human-foods in either town (●, N = 14 trials), beach (■, N = 15 trials), or remote (▲, N = 10 trials) locations. A and B present the raw data, C and D present the effect sizes of the differences between each flavour and the neutral pastry, or between town, beach, and remote locations, computed from generalised linear mixed effects models (see Methods for more details). Effects that were significantly different from zero (dashed pink vertical line) are indicated by asterisks (* $0.01 < p < 0.05$, ** $0.001 < p < 0.01$).

Figures

Figure 1.

Figure 2.

Appendix B**ROYAL SOCIETY
OPEN SCIENCE****Darwin's finches can have taste preferences, but not for
human foods**

Journal:	Royal Society Open Science
Manuscript ID	RSOS-211198
Article Type:	Research
Date Submitted by the Author:	19-Jul-2021
Complete List of Authors:	Lever, David; University of Cambridge, Zoology Rush, Louise; Laurentian University, Department of Geology Thorogood, Rose; University of Helsinki, HiLIFE; Helsingin Yliopisto, Faculty of Biological and Environmental Sciences; University of Cambridge, Zoology Gotanda, Kiyoko; University of Cambridge, Zoology; Université de Sherbrooke Faculté des Sciences, Biologie; Brock University Faculty of Mathematics and Science, Department of Biological Sciences
Subject:	behaviour < BIOLOGY, environmental science < CROSS-DISCIPLINARY SCIENCES
Keywords:	Geospiza fortis, Geospiza fuliginosis, foraging, human influences, urbanization, Galapagos
Subject Category:	Organismal and Evolutionary Biology

Author-supplied statements

Relevant information will appear here if provided.

Ethics

Does your article include research that required ethical approval or permits?:

Yes

Statement (if applicable):

This research was approved by the Galápagos National Park with permit PC-03-18.

Data

It is a condition of publication that data, code and materials supporting your paper are made publicly available. Does your paper present new data?:

Yes

Statement (if applicable):

Data are accessible as supplemental materials and are archived on Dryad:

https://datadryad.org/stash/share/F7irXFrrECSX71u2OnPA8qH3AeBMI_WOqTPCeFPCkeU

Dryad DOI: <https://doi.org/10.5061/dryad.dncjxm0h>

Conflict of interest

I/We declare we have no competing interests

Statement (if applicable):

CUST_STATE_CONFLICT :No data available.

Authors' contributions

This paper has multiple authors and our individual contributions were as below

Statement (if applicable):

KMG and RT designed the study, KMG and LVR collected the field data, DL analyzed the videos, RT, DL, and KMG analyzed the data, and all authors wrote and edited the manuscript.

Darwin's finches can have taste preferences, but not for human foods

Lever, D.¹, Rush, L. V.², Thorogood, R.^{1,3,4,7}, Gotanda, K. M.^{1,5,6,8*}

¹ Department of Zoology, University of Cambridge, Downing Street, Cambridge CB2 3EJ
United Kingdom

² Department of Geology, Laurentian University, 935 Ramsey Lake Rd, Sudbury, Ontario, P3E
2C6, Canada

³ Helsinki Institute of Life Science (HiLIFE), University of Helsinki, Helsinki 00014, Finland.

⁴ Research Program in Organismal and Evolutionary Biology, Faculty of Biological and
Environmental Sciences, University of Helsinki, Helsinki 00014, Finland.

⁵ Département de Biologie, Université de Sherbrooke, 2500, boul de l'Université, Sherbrooke,
Québec, J1K 2R1, Canada

⁶ Department of Biological Sciences, Brock University, 1812 Sir Isaac Brock Way, St.
Catharine's, Ontario, L2S 3A1, Canada

⁷ ORCID: 0000-0001-5010-2177

⁸ ORCID: 0000-0002-3666-0700

* To whom correspondence should be addressed: kg419@cam.ac.uk

Abstract

Urbanization is rapidly changing ecological niches. On the Galapagos Islands, Darwin's finches
consume human-introduced foods preferentially; however, it remains unclear why. Here we
presented pastry with flavour profiles typical of human foods (oily, salty, sweet) to small and
medium ground finches to test if latent taste preferences might drive selection of human foods. If
human-food flavours were consumed more than a neutral or bitter control at sites with human
foods, then we predicted tastes were acquired after experience with human foods; however, if no
site-differences were found then this would indicate latent taste preferences. Contrary to both
predictions, we found no evidence that human-food flavours were preferred compared to control
flavours. Instead, medium ground finches consumed the bitter control pastry most and wiped
their beaks more frequently after feeding on oily and sweet pastry (post-ingestion beak wiping
can indicate aversions). Small ground finches showed no differences in consumption but wiped
their beaks most after feeding on sweet pastry. Our results suggest that unlike many species,
medium and small ground finches do not find bitter-tasting food aversive. Furthermore, taste
preferences are unlikely to play a major role in Darwin's finches adaptation to the presence of
human foods during increased urbanization.

Key words: *Geospiza fortis*, *Geospiza fuliginosis*, foraging, human influences, urbanization,
Galapagos

Introduction

Human behaviour is now recognised to be a strong driver of local adaptation and differences
among populations of animals [1,2]. Urbanization [3–5], for example, can have profound effects
on foraging because humans often introduce novel foods to the surrounding environment either
intentionally (e.g. via garden bird feeding [6,7]) or unintentionally (e.g. by planting ornamental
plants [8,9]). This changes the diversity and availability of food items, and generates different
foraging landscapes from those in which most animals evolved [10–12]. However, organism's
responses to these altered niches vary, with some birds, for example, not only adapting more
readily to incorporate human-foods into their diet, but even preferentially consuming them over
more natural food sources [11,13–15]. These differences in if and how populations and species
utilise human foods can then have consequences for local adaptation [7,16], potentially affect
species divergence [17], or mediate the potential for invasive species to establish [18,19].

[revised manuscript text omitted]

minutes. All trials were performed from February 20th to March 25th, 2018, between 6am and
11am, or 3pm and 6pm and were filmed using a video camera (Sony HDR-CX625 Full HD
Compact Camcorder or Canon 7D Mark II with 100-400mm lens) positioned 10 metres from the
cafeteria plate. The majority of individuals were not uniquely identifiable, so we cannot be sure
that birds participating in different trials were independent. To reduce the potential for pseudo-
replication between trials, we moved the location of the cafeteria experiment for each trial by at
least 100 m within the study locations.

Each cup was filled with 2.5 g of pastry made from flour, unsalted butter, and water,
following methods from Speed and colleagues (2000). The pastry was flavoured according to
commonly available human-foods in the environment [11] and each cup was coloured (blue,
green, pink, purple, and yellow) to facilitate recognition of the contents: (i) blue indicated high in
fat (6g vegetable oil/pastry batch), (ii) green indicated bitter (0.1g quinine/pastry batch), (iii)
purple indicated sweet (23g sugar/pastry batch), (iv) yellow indicated salty (1.333g salt/pastry

batch), and (v) pink indicated neutral or unflavoured pastry. To habituate the birds to the
experimental set-up, we first conducted trials at each site with only unflavoured pastry (remote =
17 trials, beach = 17 trials, town = 19 trials; Supplemental Materials, Supplemental Table 1).
Birds can have latent colour preferences, either from experience or evolutionary history.
However, we detected no strong biases within finch species towards, or against, any of the
coloured cups based on these trials (Supplemental Materials; Supplemental Table 2;
Supplemental Figures 2 & 3).

*Video analysis*

Videos were analysed using BORIS (Behavioral Observation Research Interactive Software)
(Friard and Gamba, 2016) and each 10-minute trial was analysed by one observer (DL). As most
birds (fewer than 4%) were not individually identifiable, we counted the number of feeding
events at the level of the trial and assigned these to each species of finch based on their body size
and beak morphology. We defined a feeding event as when a bird's beak was submerged into a
cup, lifted, and then food was consumed. If at least one cup was visited during a trial, any cups
not visited received a score of 0. Following each feeding event, we then recorded the number of
152 times the finch wiped its beak on a surface within 20 seconds in accordance with published
methods on beak wiping [54,55].

46 47 155 *Statistical analyses*

Statistical analyses were undertaken using the R environment version 4.0.2 (R Core team, 2020;
analyses code and data are available as Supplemental Materials). To analyse differences in taste
preferences, we used generalised linear mixed-effect models (GLMMs) with a negative binomial

error distribution (using the `glmer.nb()` function in the `lme4` package; Bates et al. 2015) to
account for overdispersion in the number of feeding events (response variable). Trial number
was included as a random effect to account for non-independence of feeding events within trials
and the fixed effects were site, pastry flavour, and their interaction. If the interaction did not
contribute significantly to model fit (using a likelihood ratio test to compare with a simpler
model containing only additive fixed effects) it was removed, but site and pastry flavour were
retained in all models. We then used z-tests to assess the significance of differences in
consumption among pastry flavour and site using ‘neutral flavour (pink)’ and ‘town’ as the
reference level (i.e. intercept). We report estimates and standard errors and provide incidence
rate or odds ratios to compare effects.

The number of beak wipes following a feeding event were low (5 or fewer) and highly
right-skewed (medium ground finch = 3.69, small ground finch = 3.44; calculated using the
‘moments’ package; Komsta and Novomestky 2015) so we therefore we modelled the
occurrences of beak wipes using a binomial distribution, where the denominator in the response
variable was the number of feeding events when no beak wipes occurred. Assumptions of
homogeneity of variance and uniformity of the residuals for all models were checked using
Kolmogorov-Smirnov tests for uniformity, simulation tests for dispersion, and a binomial test for
outliers (implemented using the ‘DHARMA’ package [58]). The cactus finch and small tree finch
rarely came to the experimental trays so only medium and small ground finches were included in
the data set. Furthermore, attendance of medium ground finches at beach trials was very low and
they did not feed from all cups in any trial. Therefore, data from this location were excluded for
medium ground finches and the two species were analysed separately.

**Results**

A total of 53 taste preference trials were conducted across three sites. Both medium and
small ground finches varied in their propensity to participate in experimental trials at different
locations. Medium ground finches participated at 16 of the 16 trials conducted at remote sites, 0
of the 15 trials conducted at beach sites, and in 10 of the 18 trials conducted at town sites. Small
ground finches participated at 10 of the 16 trials conducted at remote sites, 15 of the 15 trials
conducted at beach sites, and 14 of the 18 trials conducted at town sites (Supplemental Table 1).

*Medium ground finches*

Medium ground finches engaged in significantly more feeding events at the remote site
than the town site (Table 1, Figure 1C, Supplemental Table 1b), but we found no evidence that
taste preferences differed according to location (location*flavour-type: $\chi^2 = 4.687$, d.f. = 4, $p =$
0.321). Medium ground finches did, however, differ in their feeding events with respect to
flavour-type (flavour-type: $\chi^2 = 11.810$, d.f. = 4, $p = 0.019$), with significantly more feeds
occurring on the bitter (green) pastry than the neutral flavoured pastry (Table 1; Figure 1A &
1C). No other flavour-types were consumed more frequently than the neutral pastry (Table 1;
Figure 1A & 1C). We next assessed behavioural wiping responses to each flavour-type (Figure
1B & 1D). However, as there were fewer than 10 trials at town sites where each flavour was
consumed (neutral (pink): $N = 5$, oily (blue): $N = 5$, bitter (green): $N = 8$, sweet (purple): $N = 3$,
salty (yellow): $N = 6$), we could not test whether responses to specific flavour-types differed
among the locations. Overall, medium ground finches were more likely to wipe their beaks after
consuming oily (blue; 14.7% of feeding events) or sweet (purple; 15.5% of feeding events)
flavours than after consuming neutral flavoured pastry (pink; Table 1; Figure 1B & 1D) and

205 wiped their beaks the least after consuming bitter (green; 5% less than after consuming neutral
pastry (Figure 1D)). Beak wiping also tended to occur more often after feeding events at town
sites compared to remote sites, although this was not significant (Table 1, Figure 1B & 1D).
Together, these results suggest that medium ground finches consumed bitter-flavoured pastry the
most (at both town and remote sites) yet wiped their beaks the least after feeding on this flavour.

17 211 *Small ground finches*

Small ground finches, on the other hand, showed no significant differences in their feeding
preferences (Figure 2), either among flavour-types ($\chi^2 = 1.160$, d.f. = 4, $p = 0.885$, Table 1;
Figure 2A & 2C), among the three sites ($\chi^2 = 2.866$, d.f. = 2, $p = 0.239$; town vs. beach estimate
= 0.424 ± 0.319 , $z = 1.329$, $p = 0.184$; town vs. remote estimate = -0.112 ± 0.357 , $z = -0.315$, $p =$
0.753 ; Table 1; Figure 2A & 2C), or in interaction ($\chi^2 = 4.728$, d.f. = 8, $p = 0.786$; Table 1;
Figure 2A & 2C). Overall, while there were no differences among locations in the proportion of
feeding events that led to beak wiping ($\chi^2 = 2.298$, d.f. = 2, $p = 0.317$; Table 1; Figure 2B & 2D),
small ground finches were 63% more likely to wipe their beaks after consuming sweet-flavoured
(purple) rather than neutral pastry (pink; Figure 2D). There were no significant differences
between other flavours and neutral-flavoured pastry (Table 1; Figure 2B & 2D). The number of
feeding trials where each flavour was consumed were too few to statistically test for location-
specific differences in beak-wiping responses to flavours. Nevertheless, together the feeding
events and beak wiping data suggest that small ground finches show little discrimination of foods
based on human-food flavours, regardless of location.

**Discussion**

Taste preferences have often been overlooked in understanding animals' foraging decisions, yet
in human-modified environments, latent taste preferences could explain why some species are
able to readily adapt to novel foods while others do not. Here we investigated if taste preferences
can explain preferential consumption of human foods by Darwin's finches [11]. We predicted
that if finches at sites with more exposure to human foods (i.e. at the tourist beach or in town)
showed greater consumption and reduced aversive behavioural responses to flavours typical of
these foods (salty, oily, sweet), then taste preferences could have developed from experience
with the changed foraging landscape. However, if finches across sites preferred these 'human-
food flavours' then latent taste preferences could have facilitated rapid adoption of human foods
into the diet. Against both predictions, the only evidence we detected for a taste preference was
that medium ground finches favoured the bitter-flavoured control. Medium ground finches fed
most often on the bitter-flavoured pastry and wiped their beaks the least following its ingestion.
This was surprising, given that bitter-tasting foods often elicit increased beak wiping in birds
[25,54]. It is possible that our sample sizes were too small to detect preferences for human-food
flavours, or that we did not add sufficient flavour to the pastry to be detectable. However,
previous work detecting taste preferences had smaller sample sizes than our study ([e.g. 31 (n =
6/group), 32 (n = 11), 33 (n = 6 and 10)], and the amount of flavour we added to the pastry
emulated human foods as closely as possible. It therefore seems unlikely that our results can only
be explained by methodological issues. Since Darwin's finches consume human foods
preferentially when available, why did we not find preferences for flavours associated with
commonly available human foods?

It could be that Darwin's finches have not evolved a preference for tastes associated with
human foods because these species are generalist feeders, especially during periods of non-
drought [10]. Taste preferences evolve when they allow animals to identify food items that offer
important nutrients (e.g., high in lipids, salts, or sugars)[25,26,29,40]. Yet for generalists, it
might not be adaptive to have latent taste preferences if these limit individuals from consuming a
wide variety of dietary items [10] or they might not need to discern specific foods that are high in
lipids, salts, or sugars. Indeed, many studies that have found taste preferences in birds have been
conducted with specialists [32,33]. Another possibility is that Darwin's finches have not yet
acquired preferences for flavours associated with human foods. At the tourist beach site, easy
access to the public only became available around 2010 (J. Podos, personal communication), and
the town of Puerto Ayora was established in 1926 [59; E. Hennessey, personal communication],
so perhaps not enough generations have passed from when finches gained access to human foods
for finches to acquire taste preferences.

For the medium ground finch, we found that the bitter taste is preferred across sites
(Figure 1; Table 1), suggesting a latent taste preference for bitter, and the presence or absence of
human foods did not correlate with different taste preferences as predicted. We also found a lack
of aversion to bitter pastry in both species with finches not wiping their bills more after
consuming bitter pastry. This was unexpected because we know birds possess TAS2R bitter taste
receptors [28], and often discriminate against toxic prey via bitterness. In fact, of the species
studied by Wang and Zhao [28], medium ground finches had the second most TAS2R genes. So
why would medium ground finches prefer bitter tastes as opposed to being averse? If the natural
foods found at remote sites are bitter in taste, that might explain why medium ground finches

have a latent preference for bitter tastes. Bitter tastes are often associated with aposematic prey,
and the relative lack of aposematic prey on the Galapagos [60] suggests finches do not need to
develop an aversion to bitter tastes as shown by the preferential consumption of bitter tastes and
not wiping their bills after consuming bitter pastry. Another possibility is zoopharmacognosy,
where animals eat medicinally advantageous foods, despite possible aversive qualities. This is
common in birds; great bustards (*Otis tarda*) ingest toxic blister beetles to control digestive tract
parasites [61] , and house sparrows (*Passer domesticus*) ingest leaves containing quinine (our
bittering agent) during malaria outbreaks [62], alleviating symptoms. Quinine is an invasive
plant found on the Galapagos. However, no finch has ever been observed consuming quinine
(Heinke Jäger, personal communication), so this possibility is unlikely.

The Galapagos Islands are experiencing an exponential increase in urbanization and
tourism, including permanent human residents [63], and we know Darwin's finches
preferentially consume human foods over natural food sources when readily available [11].
However, here we found no taste preferences for flavours associated with human foods. It
therefore seems likely that finches do not have latent preferences for these flavours, nor acquired
a preference through repeated exposure to human foods. Why then have Darwin's finches
adapted rapidly to changing food availability and incorporated human foods into their diet? One
possibility is that they could be attracted to other sensory cues such as aural or visual cues
associated with human foods. For example, in town and on the beach (but not in remote areas),
finches respond to brightly coloured visual cues of human food packaging and are attracted to
the 'crinkle' sound associated with foil and plastic food packaging [Supplemental Figure 4; 11].
Alternatively, it could be driven by availability itself at the beach and town sites. While the food

sources Darwin's finches normally feed upon are available within town and at the beach [11], the
abundance of human foods at these sites simply make these more accessible to finches, and
therefore, finches did not need to discriminate between different flavours typically associated
with human foods and aversive flavours to expand their diet diversity [11]. Further work is
required to understand the mechanisms underlying how Darwin's finches developed a preference
for consuming human foods at sites where human foods are readily available.

Humans, through processes such as urbanization, can have a major impact on foraging ecology
by introducing novel foods that can become preferentially consumed by birds [13,14]. However,
the mechanisms leading to changes in foraging ecology remain largely unknown. Although we
cannot yet explain *why* Darwin's finches prefer human foods, our results help to rule out the
possibility that taste preferences play an important role in incorporating human foods into their
diets. Similarly, our finding that ground finches do not find bitter tastes aversive expands the
increasing knowledge on variation in response to tastes among species. As the adoption of
human foods into animals' diets can have cascading effects on health, reproduction, and fitness
[6,12,16,41,43], it remains of paramount importance to elucidate why some species integrate
these foods while others do not.

44 314 **Author contributions**

KMG and RT designed the study, KMG and LVR collected the field data, DL analyzed the
videos, RT, DL, and KMG analyzed the data, and all authors wrote and edited the manuscript.

317

[revised manuscript text omitted]

 finches from coloured cups containing pastry flavoured to be salty (yellow), sweet (purple),
 bitter (green), or oily (blue). Differences in (I) were estimated using generalised linear mixed
 effects models with a negative binomial error distribution and where trial and site were included
 as random effects. Differences in (II) were estimated using a similar model but with a binomial
 error distribution to account for proportional response data. ‘Neutral’, unflavoured pastry was set
 as the intercept (in parentheses) in all models.

	I. Feeding events			II. Beak wiping		
	Mean difference \pm S.E.	Z	p	Mean difference \pm S.E.	Z	p
(a) Medium ground finches:						
(neutral)	0.689 \pm 0.367	1.879	0.060	-2.696 \pm 0.345	-7.489	<0.001
Oily (blue)	-0.442 \pm 0.367	-1.206	0.228	0.905 \pm 0.433	2.090	0.037
Bitter (green)	0.751 \pm 0.355	2.118	0.034	-0.053 \pm 0.403	-0.131	0.896
Sweet (purple)	0.427 \pm 0.353	1.207	0.227	0.936 \pm 0.368	2.541	0.011
Salty (yellow)	0.207 \pm 0.355	0.584	0.559	0.593 \pm 0.399	1.486	0.137
Site - remote	1.075 \pm 0.359	2.993	0.003	0.565 \pm 0.320	1.765	0.078
(b) Small ground finches:						
(neutral)	1.827 \pm 0.348	5.258	<0.001	-1.815 \pm 0.257	-7.050	<0.001
Oily (blue)	-0.326 \pm 0.428	-0.762	0.446	0.060 \pm 0.276	0.217	0.828
Bitter (green)	-0.152 \pm 0.427	-0.356	0.722	-0.024 \pm 0.274	-0.087	0.931
Sweet (purple)	-0.092 \pm 0.425	-0.215	0.830	0.489 \pm 0.242	2.017	0.044
Salty (yellow)	-0.396 \pm 0.427	-0.929	0.353	0.121 \pm 0.271	0.448	0.654
Location	0.424 \pm 0.319	1.329	0.184	0.144 \pm 0.255	0.563	0.573
	-0.112 \pm 0.357	-0.315	0.753	0.438 \pm 0.291	1.508	0.132

Figure Legend

Figure 1. Differences in the (A,C) number of feeding events and (B,D) proportion of feeding
events that were followed by beak-wiping by Medium ground finches presented with pastry in
coloured cups with neutral (pink), oily (blue), bitter (green), sweet (purple), or salty (yellow)
flavours common to human-foods in either town (●, N = 10 trials) or remote (▲, N = 16 trials)
locations. A and B present the raw data, C and D present the effect sizes of the differences
between each flavour and the neutral pastry, or between town and remote locations, computed
from generalised linear mixed effects models (see Methods for more details). Effects that were
significantly different from zero (dashed pink vertical line) are indicated by asterisks (* 0.01 < p
< 0.05, ** 0.001 < p < 0.01).

Figure 2. Differences in the (A,C) number of feeding events and (B,D) proportion of feeding
events that were followed by beak-wiping by small ground finches presented with pastry in
coloured cups with neutral (pink), oil (blue), bitter (green), sweet (purple), or salty (yellow)
flavours common to human-foods in either town (●, N = 14 trials), beach (■, N = 15 trials), or
remote (▲, N = 10 trials) locations. A and B present the raw data, C and D present the effect
sizes of the differences between each flavour and the neutral pastry, or between town, beach, and
remote locations, computed from generalised linear mixed effects models (see Methods for more
details). Effects that were significantly different from zero (dashed pink vertical line) are
indicated by asterisks (* 0.01 < p < 0.05, ** 0.001 < p < 0.01).

**Figures**

Figure 1.

Figure 2.

Appendix C

October 26, 2021

Dear Dr. Lukas and Mr. Padian,

Many thanks to you, and the reviewers for thoughtful comments and queries on our manuscript, “Darwin’s finches can have taste preferences, but not for human foods” submitted to the Royal Society Open Science. We appreciate the opportunity to revise our manuscript by considering the suggestions for improvement. Below, we respond in italics to each of the comments and explain how we have resolved or addressed the question/issue. I hope this revised manuscript is now suitable for publication in Royal Society Open Science. If you have any further questions or suggestions for improvement, we will be happy to address them.

Sincerely,
Dr. Kiyoko Gotanda
David Lever
Louis Rush
Dr. Rose Thorogood

Associate Editor Comments to Author (Dr Dieter Lukas):

Associate Editor: 1

Comments to the Author:

Dear authors,

Your article entitled “Darwin’s finches can have taste preferences, but not for human foods” has now been seen by two reviewers and the reviewers’ comments are appended below. As you will see, both reviewers consider your study to contribute relevant insights into how sensory information might relate to avian adaptation to urbanisation, and I share their views. Yet they have several comments that need to be addressed carefully before I would consider recommending this article for publication.

We thank you and the reviewers for your thoughtful comments. We have carefully considered each comment and the suggestions have greatly improved the manuscript. We address each of the individual comments below.

The main issues raised are about the limitations of this study. Studies such as these on wild birds are difficult to plan and conduct, so any additional information gained can be helpful. However, the limitations should be acknowledged and the potential implications of the results tempered accordingly.

We have edited the manuscript to better reflect the limitations of our study, which we expand upon below.

I agree with reviewer 2 that the small sample sizes, due to birds often not participating in the experiments, limit the ability to draw firm conclusions about the behaviour under investigation. You acknowledge this limitation in the discussion, but come to the conclusion that it "seems unlikely that our results can only be explained by methodological issues" (line 245f). You base this conclusion on the observation that some studies with small sample sizes did find effects.

However, there might well be publication bias (as has been shown for essentially every other research question), with small studies that did not find an effect underreported. With small sample sizes, power is low to detect effect, which could mean that you are in the proportion of cases where you did not find an effect even though it is there. This appears further confounded by the fact that your sample sizes might be even smaller than what you report. For one, you mention that you "cannot be sure that birds participating in different trials were independent", and in addition you also might have had social effects that mean the observations are not fully independent. I would therefore ask you to edit your manuscript to more clearly state your findings in the lights of this limitation. One change would be to add the actual sample sizes to each reported result (after every p-value). The other would be to make less definite statement about the implications of what you found. For example, the final two sentences of the abstract state that you found support for the hypotheses that ground finches "do not find bitter-tasting food aversive" and that "taste preferences are unlikely to play a major role". Phrasing it in terms of saying that you found no evidence that ground finches find bitter-tasting food aversive and that taste preferences are linked to the consumption of human food can make it clearer that these are the conclusions within the limitations of your study rather than a general statement about the species.

We have added sample sizes in the table legends and we have modified the abstract and discussion to better reflect the limitations.

In addition, I agree with reviewer 1 that also other limitations besides sample size, such as seasonality or lack of control on availability of native foodstuffs, should be more clearly acknowledged. This again limits the power you have to draw definite conclusions, also supporting the changes to the statements of the implications of your study.

We detail our responses to Reviewer #1's comments below.

In addition to these main issues, both reviewers also have provided annotations on your manuscript with further more specific comments on particular sentences.

We copied the comments below and then address each one individually.

All the comments focus on changing the presentation of results, changing the phrasing in places and adding information to the manuscript that either presumably was already part of your decision processes or can be added through further further references. Addressing the comments appears feasible, and I am looking forward to read a new improved version of the manuscript.

We thank the editor for this opportunity and look forward to your comments and decision.

Reviewer comments to Author:

Reviewer: 1

Comments to the Author(s) - see also attachment 'Colour and Taste Preferences Lever Manuscript'.

This study represents an interesting addition to the growing literature on the behavioural

responses of fauna to urbanisation. The experimental setup and analyses are adequate, and the presentation of results is clear. The discussion is appropriate. I have made a few suggestions in the manuscript (see Word document attached) that I hope are useful. I would like to recommend to the authors to review the available literature on scrounging behaviour of Kea in NZ of seagulls in Australia and NZ as these are stark examples of behavioural changes in response to junk food availability.

We now reference the keas that are able to open rubbish bins to access the food inside and the gulls who consume human food waste.

The authors should acknowledge explicitly other limitations of their study, such as seasonality, and lack of control on availability of native foodstuffs.

We now acknowledge these other limitations in the discussion

Overall a neat study.

Thank you!

Title: The title is not very informative, as the study only pertains a couple of finch species at a single location. I would suggest changing the title to either a) question format or b) specific to the system studied.

We have modified the title as follows to include the species and better reflect our results: Darwin's small and medium ground finches might have taste preferences, but not for human foods.

L3: Only on human-inhabited islands (4-5?) which represent less than 5% of the land of the archipelago...please be specific about this.

We now say "On the inhabited Galapagos Islands..."

L3: I would say "this has received little attention" or "has received growing attention in recent years"...after all your study doesn't address this question either.

We agree but we are not sure either of those phrases are quite right. We think we should leave it as is because we do elaborate in the introduction the link between taste preferences and preferential consumption of human foods.

L3: three

We believe the reviewer is referring to the three tastes we list in this sentence, though we had a total of five flavours, so we think keeping the wording is the most descriptive for the abstract. We essentially did have three flavours (salt, sweet, oil) and the other two (neutral, bitter) were treated more as controls.

L16: Could it simply be “availability”? I would be cautious about the use of the term “adaptation” to the presence of human foods as this would suggest some physio-ecological changes, which were not directly assessed in this study.

Yes, we discuss how the preference for human foods could be due to availability of human foods in urban areas. We now talk more about responses that could be adaptive as opposed to adaptation itself as the reviewer is correct in that we cannot say it's adaptation without knowing if genetic changes have occurred as well.

L26: What about introduced species? This could also be a source of non-native foodstuffs (i.e. insects, carrion, etc.).

We agree that other species can be introduced and become novel food source. We now state: “Urbanization [3–5], for example, can have profound effects on foraging because humans often introduce novel foods to the surrounding environment either intentionally (e.g. via garden bird feeding [6,7]) or unintentionally (e.g. by planting ornamental, invasive plants [8,9] or food waste [10,11]).”

L30: I would suggest rephrasing this to “native food sources”. i.e. an introduced ornamental fruit is as natural as a native one!

We have changed it to “native food sources”

L36: There are a number of studies on the Kea (a New Zealand parrot) and its scrounging behaviour, as well as the near fatal consequences after ingesting junk food, perhaps add this as an extreme example of other consequences of changing foraging landscapes.

We thank the reviewer for these examples and now include them: “Taste preferences can also have implications in conservation, mediating the potential for invasive species to establish [22,23], for example, or making native species vulnerable to accidental poisoning [24].”

L47: potentially

We have changed ‘presumably’ to ‘potentially’.

L54: Add examples of Kea as well as seagulls in Australia and NZ.

Done!

L63: This could have serious consequences for biological conservation also, see notes re: kea above.

We now allude to the kea and gull example.

L71: Living in. or near urban. Areas???

Since we have our remote site, we cannot say living in urban areas. We have clarified it to be specific to islands with permanent human populations. "Here, we investigate if Darwin's finches, which on human-inhabited islands are known to vary in their preferential consumption of human foods."

L79: Again...only applicable to humaninhabited islands...this statement reads as if all (13-14?) Finch species in the Galapagos exhibit this trait, which is not the case.

We have changes this to "Indeed, we know that Darwin's finches on human inhabited islands can preferentially consume human foods..."

L83: Not only beaches tho. I would say "tourist areas"...there are areas in the highlands of San Cristobal and Santa Cruz where numerous sgf forage on tourist food.

We have changed to 'tourist areas'. On a side note, we find it interesting finches on San Cristobal have an affinity for butter (KMG personal observation).

L100: Beak wiping behaviour or beak-wiping behaviour?

We have changed all mentions of this to the hyphenated version.

L105: There are about eight species of Darwin's finches on Santa Cruz, be specific about the species studied.

*We detail the focal species in our third sentence: "The two focal species were small ground finches (*Geospiza fuliginosa*) and medium ground finches (*Geospiza fortis*)."*

L108: Local name?

We have added the local name of El Garrapatero proper to not confuse it with El Garrapatero beach.

L108: Tourist site? Also no need for " "

We have removed the quotes and clarified the beach site is a tourist site

L109: How many km from main urban town?????

We have added the information "the beach site was El Garrapatero beach, a tourist, non-urban site 12 km from Puerta Ayora where visitors often bring picnics so human food is present and abundant..."

L111: Where specifically? A restaurant? A park? Near the beach? Again be specific.

We have now specified it is throughout town: "...and the town site was Puerto Ayora, a fully urbanized town where humans and their food are ubiquitous throughout the entire town."

L113: Which genus?

We now list the species: "...two other finch species (the cactus finch, Geospiza scandens and the small tree finch, Camarhynchus parvulus were also present occasionally, but rarely interacted with our experiments."

L117: Conducted?

We have changed 'performed' to 'conducted'.

L119: Egg of what? Emu? Ostrich? Chicken? Is there a photograph of the experimental setup that can be included as supplementary info? Approx. volume or capacity might be more informative.

We now say "Five cups the size of ½ a plastic egg the size of a chicken egg..." Unfortunately, we do not have access to the egg cups because they are stored in the Galapagos and due to the pandemic have not been able to access them. Furthermore, it is currently not close to Easter which is when you can usually purchase them. We hesitate to estimate the diameter because we do not want to mislead readers.

L120: Circumference? I think it is important to be very detailed about the experimental setup for replication purposes. If this info is available in other published studies be specific about this.

Again, the plates were left in the field and we do not have access. We have added "on a white plastic dinner plate that encircled the tray (e.g. the tray did not hang over the sides)"

L123: Is this the wet or dry season in the Galapagos islands? Potential seasonal effect?

Unfortunately, with climate change, we don't know anymore. Technically it's during the wet season but KMG and LVR do not recall it raining very much while in the field. However, we have added: "All trials were performed from February 20th to March 25th, 2018, which is traditionally in the rainy season between..."

L129: Perhaps a map as supplementary info???

We have included a map in the supplementary material as Supplemental Figure 1.

L134: Mmm %? How many g is a pastry batch? I am aware this information might be detailed in the reference provided but this info can easily be included here.

We now include the recipe as follows: "The pastry (335 g flour, 135 g unsalted butter, and 30 g water) was flavoured according to commonly available human-foods..."

L141: I might be wrong, but I believe that flowers and fruits in the Galapagos Islands have a limited colour palette....are there studies exploring the response of finches to different colours?

Many of the flowers of endemic Galapagos plants are yellow due to their co-evolution with the carpenter bee, one of the main pollinators on the Galapagos. We did run trials with only neutral colours, and yellow was preferred a bit in one species at one site, which is not indicative of an overall preference for yellow. One thesis has looked at colour preferences of food and they found an aversion to blue, but our food was not coloured, just the cup, and we did not find such an aversion. We are not aware of other colour preference trials done with Darwin's finches.

L149: I understand that distinguishing between species of Darwin's finches can be quite challenging...could you indicate to the reader what are the distinguishing features of each species? Perhaps illustrate with photos the differences between species?

We now include the following: "Species identification was done based on bill and body size in comparison to the experimental trays. The observer was trained by KMG by first observing and identifying stills of the video, and then with actual clips."

L177: Scientific names?

The scientific names are now included above at their first mention.

L246: Perhaps make a note about potential seasonal effects? Could it be that finches exhibit a stronger preference during the dry vs wet season?

Yes, we have now added: "We conducted our trials during the traditional rainy season when the finches would be more generalist."

L251: Ok here be specific about the conditions during your study. Was it raining heavily? Drought?

Please see our previous response.

L259: Not quite sure about this one....I recall access to El Garrapatero was available prior to this date

It was accessible, but for a long time, there was a locked gate that you needed a ranger to open for you. Even once the gate was removed, it was still difficult to access the beach. It was not until the new, current parking lot was built that the beach could be regularly accessed, and we're pretty sure the new parking lot was built in 2010. If you have more accurate information when the new parking lot was built, please do let us know and we will correct the manuscript.

L262: Perhaps a more informative landmark would be growth in tourism? In 2008-2015 tourism to the Galapagos islands grew very rapidly i.e. more opportunities for finches to interact with humans at a variety of sites

The reviewer makes an interesting point. We have added the relatively recent exponential increase in tourism: "Tourism has grown exponentially only relatively recently [59]."

L287: At least in the season during which the experiments were conducted

We have revised this to: "However, here we found no evidence for taste preferences for flavours associated with human foods during the traditional rainy season."

L290: Again only at urban sites. Could it be availability? i.e. search for scarce native seeds for one hour vs land on a pile of rice?

We have revised the sentence: "Why then have Darwin's finches adapted rapidly to changing food availability and incorporated human foods into their diet when human foods are readily available?"

L295: Yes!

We are happy reviewer agrees with this statement

L296: Yeah but this study did not control for availability of natural foods, correct?

The reviewer is correct, and we have revised: "While the food sources Darwin's finches normally feed upon are available within town and at the beach [11] (though we did not control for this), the abundance of human foods at these sites simply make these more accessible to finches, and therefore, finches did not need to discriminate between different flavours typically associated with human foods and aversive flavours to expand their diet diversity [11]."

L304: Examples from other organisms? I understand the focus here is birds, but perhaps worth mentioning non-avian examples?

We have intentionally tried to limit our discussion to birds given the specificity of our study.

Reviewer: 2

Comments to the Author(s) - see also attachment 'RSOS-211198_Proof_hi'.

I have a problem with the whole manuscript. The authors focused on really important and interesting problem of food taste choice, but design really does not provides final solution. I put my comments also on the MS. I also suggest some changes in statistics.

We thank the reviewer for their comments. We think that our wording about conclusions about taste preferences (or not) might have been too conclusive based on the experiment. We believe that by tempering our statements to be more about evidence for or lack of taste preferences as opposed to conclusions about taste preferences will help resolve the link between the experiment and the conclusions. We are unsure what additional statistics should be employed for this manuscript. We feel we have properly accounted for all of our variables including appropriate random effects and design.

L26: but also via food, by waste - see for example: Tryjanowski, P., Skórka, P., Sparks, T. H., Biaduń, W., Brauze, T., Hetmański, T., ... & Wysocki, D. (2015). Urban and rural habitats differ in number and type of bird feeders and in bird species consuming supplementary food. *Environmental Science and Pollution Research*, 22(19), 15097-15103.

We now include this as an example: “[1,2]. Urbanization [3–5], for example, can have profound effects on foraging because humans often introduce novel foods to the surrounding environment either intentionally (e.g. via garden bird feeding [6–8]) or unintentionally (e.g. by planting ornamental, invasive plants [9,10] or food waste [11,12]).”

L30: and it has some not only behavioural but medical consequences: Tryjanowski, P., Møller, A. P., Morelli, F., Indykiewicz, P., Zduniak, P., & Myczko, Ł. (2018). Food preferences by birds using bird-feeders in winter: a large-scale experiment. *Avian Research*, 9(1), 1-6.

We thank the reviewer for this reference and now cite the study

L201: I still have a problem how to divide colour from test? both arranged different neurons

We had done trials focusing on colour. We agree that colour and taste are linked because we wanted the finches to be able to have something to identify the different tastes with. We also performed trials before the taste trials to ensure that there were no colour preferences. We have reworded the manuscript to make this more clear.

L235: not really if the first stimuli is vision... sorry I have a problem with understanding the design

We know Darwin's finches preferentially consume human foods. Previous work did experiments where human foods and foods the finches are suppose to consume were presented on a tray. There was no visual colour associated with the foods in this experiment. Furthermore, this sentence is in regards to finches developing possibly developing taste preferences and is not in reference to any experiments.

L243: true

We are glad the reviewer agrees with our statement.

L327: check formats; sometimes bolded, sometimes not; abbreviations in non-consequent way

We have cleaned up the references. The bolding of issue is a journal formatting requirement for Royal Society journals.

Appendix D

December 09, 2021

Dear Dr. Lukas and Mr. Padian,

Thank you again for your comments and queries on our revision “Darwin’s finches can have taste preferences, but not for human foods” re-submitted to Royal Society Open Science. Again, we appreciate the opportunity to revise our manuscript. Below, we respond in italics to each of the comments and explain how we have resolved or addressed the question/issue. We hope this revised manuscript is now suitable for publication in Royal Society Open Science. If you have any further questions or suggestions for improvement, we will be happy to address them.

Sincerely,
Dr. Kiyoko Gotanda
David Lever
Louis Rush
Dr. Rose Thorogood

Associate Editor Comments to Author (Dr Dieter Lukas):
Associate Editor
Comments to the Author:

Dear authors,

Thank you for replying to the reviewers' and my comments in such detail, and for changing the phrasing in your manuscript to reflect the potential limitations.

Thank you for the opportunity!

I have a follow-up to the critical comments reviewer 2 made. I realise that their initial comments were not very detailed, and I also had not been entirely sure what they were criticising. After reading your replies, I now am wondering the following about their comments about the setup and the stimuli involved.

If I understand your setup correctly, each trial was performed at a different location.

Yes, the person conducting the experiment moved around for each trial within a particular site.

Accordingly, you assume that for each bird that participates, they only see the setup once when their data is being collected. That would mean that birds coming in are naive to the tastes, and in particular to the fact that a given colour indicates a particular flavour. Doesn't that mean that for the analyses you have to exclude all zeroes? A bird not feeding from a particular cup does not tell you anything about the preference or aversion for that respective taste because that bird does not know that the taste is there. These birds cannot be said that they are avoiding that cup because of the taste, because they never experienced the taste of the pastry in that cup. So I think the only meaningful information is in what happens after a bird actually starts feeding from a given cup: do they continue or do they stop because they now have experienced that taste. That means you see whether the feeding rate of those events where a bird started feeding is higher for

some flavours than for others. I think this has to be at the level of individual birds, rather than the non-zero feeding rate during a given trial.

We now understand what Reviewer #2 had meant, and based on this explanation, we understand why including our zeros might not be the best option. We used this approach because we assumed that many birds would visit the trials repeatedly, and thus a zero was likely to indicate a choice. However, we have since checked the data again and many of the feeding bouts included feeds from only one of the cups. Therefore, we have adopted the suggested approach and now present revised results where 0s are not included in our analyses.

Excluding the 0s, however, meant that our data became very uneven across flavours and locations. We have therefore now combined the two species in our analyses and include species as a fixed effect and as a nested random effect (within trial, as both species sometimes participated in the same trials) to account for the study design.

With the new analyses, our results change but our principle finding remains – there are no strong preferences for flavours associated with human foods. However, we do now find some weak evidence for taste preferences to differ among locations (a weak interaction between location and food flavour): oily flavour was preferred less, but only at remote sites. The interaction between flavour and location is weak ($p = 0.0376$) so this result should be interpreted cautiously. However, our data on behavioural responses after tasting the foods does support the pattern: at remote sites, we find beak wiping is higher after tasting ‘human food’ flavours compared to our two controls (neutral and bitter). Interestingly, however, we still find no evidence that bitter food is aversive, either in terms of feeding events or beak wiping. We have revised the manuscript to reflect the new analyses.

In addition, how would a bird know that the pastry in a cup different from the one they started with at random would taste different? So why would you expect a bird that has started on a neutral cup to stop feeding there and switch to for example the cup with the sweet pastry? If birds are truly naive, wouldn't you expect that after a bird picks one of the cups at random it just continues eating there - which would mean you would not expect to be able to tell whether there are taste differences with this setup? If I am wrong, please add an explanation to the manuscript why you are predicting different feeding rates for the different flavours with your setup.

For a single finch, we agree – the finch has no reason to expect that the pastry in another cup would taste differently. However, it is well known that persistence with a cognitive task varies among individuals, so it is also very likely that returning individuals would vary in whether they continue to feed from the same cup or try another. Darwin's finches also commonly forage in groups, meaning that there are opportunities for individuals to observe and learn from the foraging choices of others. If the bitter taste were aversive and resulted in more beak wiping for example, other finches present could learn to avoid the bitter (green) cup. Alternatively, if a flavour was preferred then this too could spread. Our experimental design did not, however, allow us to control for social effects such as these.

Two minor comments:

I am not familiar with these species, so it was sometimes confusing to read about medium and small ground finches because I expected this to potentially reflect multiple species that are grouped by size, rather than each term referring to one specific species. Maybe you could already

in the abstract list the Latin species name when you first introduce medium and small ground finches to clarify this (line 14). Somewhat linked to that, the legend for Figure 1 in line 543 has the "Medium" capitalised whereas in the remainder of the text it is spelled in small letters.

We now clarify that, in the current literature, small and medium ground finches are considered separate species. We have clarified this in the abstract and text and have corrected the capitalization in the Figure 1 legend.

I am not sure why you removed the sample size information from the result section (lines 210, 211, 229, 234). As I mentioned before, I think it would be helpful to have the mention of the sample size throughout. Here especially, because before you only speak about trials whereas these analyses are actually on the level of feeding and beak-wiping events.

We had moved them to the table to make the results text more readable but have now added them back into the text.

Appendix E

December 17, 2021

Dear Dr. Lukas and Mr. Padian,

Thank you for accepting our revised manuscript “Darwin’s small and medium ground finches might have taste preferences, but not for human foods” at Royal Society Open Science. Below, we respond in italics to each of the comments and explain how we have resolved or addressed the question/issue. We thank you for this opportunity and look forward to seeing this manuscript published in Royal Society Open Science

Sincerely,
Dr. Kiyoko Gotanda
David Lever
Louis Rush
Dr. Rose Thorogood

Associate Editor Comments to Author (Dr Dieter Lukas):
Associate Editor
Comments to the Author:
Dear authors,

Thank you for thinking the issues around the design through, and for adjusting the analyses accordingly. I think there are now some insightful findings about the behaviour of these finches around human food flavours!

Thank you again and we also pleased with the accepted manuscript!

I only have three remaining comments regarding your implementation of the revisions:

1) Abstract, line 18ff: "Our results suggest, therefore, that while Darwin’s finches may have acquired a tolerance to the flavours of human food"

I wonder whether you want to specify that the "Darwin's finches regularly exposed to human food might have acquired a tolerance to those flavours". You did detect a difference between the town compared to the remote site.

Changed.

2) Discussion, line 302ff: "This suggests that ground finches do not have latent taste preferences for human foods, or have acquired preferences from contact with human foods" The second part, that birds in the site with more human food have no acquired differences, seems to go against your statement in the results ("at remote sites, all human-food flavours evoked more behavioural reactions than controls") and the sentence quoted above from the abstract. I can see that there are no preferences, but it seems like it would be good to clarify that birds at the town site seem to have lost their aversion.

We noticed that ‘or’ should be ‘nor’ and have clarified the loss of aversion. “This suggests that ground finches do not have latent taste preferences for human foods nor have acquired taste preferences from contact with human foods. Furthermore, ground finches at sites with human foods might now be more tolerant of oily flavours and lost their aversion to tastes associated

with human foods.”

3) The top panels of all the figures still show the full data, including the zeroes. I think that is fine, but it probably would be worth explaining in the methods that you excluded the zeroes and provide explanations for this approach from our exchange; and worth mentioning in the figure legend that they show the full dataset including events where birds did not feed.

We do have the plots with no 0's for the feeding data. We think the track changes might have made it confusing. In this cleaned up version (with only the most recent track change edits), you can more readily see that the plot has points for 1 or above, and not for zero. We apologize for the confusing nature of track changes making it difficult to see the appropriate plot. For the wiping data, the 0's indicate 0% of those feeds involved beak wiping.